# Fast Certified Robust Training with Short Warmup

**Zhouxing Shi[1*], Yihan Wang[1*], Huan Zhang[1,2], Jinfeng Yi[3], Cho-Jui Hsieh[1]**
[1]University of California, Los Angeles  [2]Carnegie Mellon University  [3]JD AI Research
zshi@cs.ucla.edu, yihanwang@cs.ucla.edu, huan@huan-zhang.com,
yijinfeng@jd.com, chohsieh@cs.ucla.edu
* Equal contribution

## Abstract

Recently, bound propagation based certified robust training methods have been proposed for training neural networks with certifiable robustness guarantees. Despite that state-of-the-art (SOTA) methods including interval bound propagation (IBP) and CROWN-IBP have per-batch training complexity similar to standard neural network training, they usually use a long warmup schedule with hundreds or thousands epochs to reach SOTA performance and are thus still costly. In this paper, we identify two important issues in existing methods, namely exploded bounds at initialization, and the imbalance in ReLU activation states and improve IBP training. These two issues make certified training difficult and unstable, and thereby long warmup schedules were needed in prior works. To mitigate these issues and conduct faster certified training with shorter warmup, we propose three improvements based on IBP training: 1) We derive a new weight initialization method for IBP training; 2) We propose to fully add Batch Normalization (BN) to each layer in the model, since we find BN can reduce the imbalance in ReLU activation states; 3) We also design regularization to explicitly tighten certified bounds and balance ReLU activation states during wamrup. We are able to obtain **65.03%** verified error on CIFAR-10 ($\epsilon = \frac{8}{255}$) and **82.36%** verified error on Tiny-ImageNet ($\epsilon = \frac{1}{255}$) using very short training schedules (**160 and 80 total epochs**, respectively), outperforming literature SOTA trained with hundreds or thousands epochs under the same network architecture. The code is available at `https://github.com/shizhouxing/Fast-Certified-Robust-Training`.

## 1 Introduction

While deep neural networks (DNNs) are successfully applied in various areas, its robustness problem has attracted great attention since the discovery of adversarial examples (Szegedy et al., 2013; Goodfellow et al., 2015; Carlini & Wagner, 2017; Kurakin et al., 2016; Chen et al., 2017; Madry et al., 2018; Su et al., 2018; Choi et al., 2019), which poses concerns in DNN applications especially the safety-critical ones such as autonomous driving. Methods for improving the empirical robustness of DNNs, such as adversarial training (Madry et al., 2018), provide no provable robustness guarantees, and thus some recent works aim to pursue *certified robustness*. Specifically, the robustness is evaluated in a certifiable manner using robustness verifiers (Katz et al., 2017; Zhang et al., 2018; Wong & Kolter, 2018; Singh et al., 2018, 2019; Bunel et al., 2017; Raghunathan et al., 2018b; Wang et al., 2018b; Xu et al., 2020; Wang et al., 2021), which verify whether the model is provably robust against all possible input perturbations within the range. This is achieved usually by efficiently computing the output bounds.

To improve certified robustness, *certified robust training* methods (also referred to as certified defense) minimize a certified robust loss computed by a verifier, and the certified loss is an upper bound of the worst-case loss given specified input perturbations. So far, Interval Bound Propagation (IBP) (Gowal

et al., 2018; Mirman et al., 2018) and CROWN-IBP (Zhang et al., 2020; Xu et al., 2020) are the most efficient and effective methods for general models. IBP computes an interval with the output lower and upper bounds for each neuron, and CROWN-IBP further combines IBP with tighter linear relaxation-based bounds (Zhang et al., 2018; Singh et al., 2019) during warmup.

Both IBP and CROWN-IBP with loss fusion (Xu et al., 2020) have a per-batch training time complexity similar to standard DNN training. However, certified robust training remains costly and challenging, mainly due to their unstable training behavior – they could easily diverge or stuck at a degenerate solution without a long "warmup" schedule. The warmup schedule here refers to training the model with a regular (non-robust) loss first and then gradually increasing the perturbation radius from 0 to the target value in the robust loss (some previous works also refer to it as "ramp-up"). For example, generalized CROWN-IBP in Xu et al. (2020) used 900 epochs for warmup and 2,000 epochs in total to train a convolutional model on CIFAR-10 (Krizhevsky et al., 2009).

In this paper, we identify two important issues in existing certified training, so that a long warmup schedule could not be easily removed in previous works. First, we find that the certified bounds can explode at the start of training, which is partly due to the suboptimal *weight initialization* in prior works. A good weight initialization is important for successful DNN training (Glorot & Bengio, 2010; He et al., 2015a), but prior works for certified training generally use weight initialization methods originally designed for standard DNN training, while certified training is essentially optimizing a different type of augmented network defined by robustness verification (Zhang et al., 2020). The long warmup with gradually increasing perturbation radii in prior works can somewhat be viewed as finding a better initialization for final IBP training with the target radius, but it is too costly. Second, we also observe that *IBP leads to imbalanced ReLU activation states*, where the model prefers inactive (dead) ReLU neurons significantly more than other states because inactive neurons tend to tighten IBP bounds. It can however hamper classification performance if too many neurons are dead. This issue can become more severe if the warmup schedule is shorter.

We focus on improving IBP training, since IBP is efficient per batch, and it is also the base of recent state-of-the-art methods (Zhang et al., 2020; Xu et al., 2020). We propose the following improvements:

- We derive a new weight initialization, *IBP initialization*, for IBP-based certified training. The new initialization can stabilize the tightness of certified bounds at initialization.

- We identify the benefit of Batch Normalization (BN) in certified training, and we find BN which normalizes pre-activation outputs can balance ReLU activation states and also stabilize variance. We propose to fully add BN to every layer, while it was partly or fully missed in prior works.

- We further propose regularizers to explicitly stabilize certified bounds and balance ReLU activation states during warmup.

We are able to efficiently train certifiably robust models that outperform previous SOTA performance in significantly shorter training epochs. We achieve a verified error of **65.03%** ($\epsilon = \frac{8}{255}$) on CIFAR-10 in **160** total training epochs, and **82.36%** on TinyImageNet ($\epsilon = \frac{1}{255}$) in **80** epochs, based on efficient IBP training. Under the same convolution-based architecture, we significantly reduce the total training cost by $20 \sim 60$ times compared to previous SOTA (Zhang et al., 2020; Xu et al., 2020) or concurrent work (Lyu et al., 2021).

## 2  Background and Related Work

### 2.1  Certified Robust Training

Training robust neural networks can generally be viewed as solving the following min-max optimization problem:

$$\min_\theta \mathbb{E}_{(\mathbf{x},y)\in\mathcal{X}} \left[ \max_{\delta\in\Delta(\mathbf{x})} L(f_\theta(\mathbf{x}+\delta),y) \right], \tag{1}$$

where $f_\theta$ stands for a neural network parameterized by $\theta$, $\mathcal{X}$ is the data distribution, $\mathbf{x}$ is a data example, $y$ is its ground-truth label, $\delta$ is a perturbation constrained by specification $\Delta(\mathbf{x})$, and $L$ is the loss function. Empirical *adversarial training* methods (Goodfellow et al., 2015; Madry et al., 2018) solve the inner maximization in Eq. (1) with adversarial attack, and then solve the outer

minimization as regular DNN training but augmented with $\delta$. However, in adversarial training, the inner maximization has no guarantee to find a $\delta$ which can lead to worst model performance. In contrast, *certified robust training* methods compute a certified upper bound for the inner maximization, so that the upper bound provably covers the worst-case perturbation.

In terms of certified robustness works, Raghunathan et al. (2018a) used semidefinite relaxations for small two-layer models, and Wong & Kolter (2018); Mirman et al. (2018); Dvijotham et al. (2018); Wang et al. (2018a) used linear relaxations but are still too computationally expensive for large models. On the other hand, Mirman et al. (2018) first used interval bounds to train a certifiably robust network, and Gowal et al. (2018) made it more effective. This approach is often referred to as interval bound propagation (IBP). CROWN-IBP (Zhang et al., 2020) further combined IBP with tighter linear relaxation bounds by CROWN (Zhang et al., 2018) during warmup, and it is generalized and accelerated in Xu et al. (2020). Additionally, Balunovic & Vechev (2020) combined certified training with adversarial training; Xiao et al. (2019) added a ReLU stability regularizer to empirical adversarial training, to reduce unstable neurons for faster and tighter verification when tested with mixed integer programming (MIP), but their objective is distinct from ours and this method was shown not to improve certified training (Lee et al., 2021). In concurrent works, Lyu et al. (2021) proposed a parameterized ramp function as an alternative activation function, and used a tighter linear bound propagation algorithm for verification; Zhang et al. (2021) proposed to use a different architecture with "$\ell_\infty$-distance neurons" instead of traditional linear or convolutional layers. Yet they still need long training schedules.

Moreover, while our scope in this paper is deterministic certified robustness, there are also randomization based works for probabilistic certified defense (Cohen et al., 2019; Li et al., 2019; Lecuyer et al., 2019; Salman et al., 2019). But randomized smoothing requires costly sampling at test time, and it is usually for $\ell_2$ perturbations and has fundamental limitations for $\ell_\infty$ ones (Yang et al., 2020; Blum et al., 2020; Kumar et al., 2020).

### 2.2 Weight Initialization of Neural Networks

Many prior works have studied the weight initialization for standard DNN training. Xavier or Glorot initialization (Glorot & Bengio, 2010), adopted by popular deep learning libraries such as PyTorch (Paszke et al., 2019) and Tensorflow (Abadi et al., 2016) as the default initialization, aim to stabilize the magnitude of forward propagation and gradient backpropagation signals measured with variance. It uses a uniform distribution or normal distribution to independently initialize each element in the weight matrix with a derived variance for the distribution. He et al. (2015a) derived an initialization that more accurately stabilizes the variance in ReLU networks. Saxe et al. (2013) proposed an orthogonal initialization which may lead to better learning dynamics. Some other works also derived initializations for specific DNN structures (Taki, 2017; Huang et al., 2020), and Bhattacharya (2020); Zhu et al. (2021) proposed to automatically learning initializations. However, these initializations were designed for standard DNN training, while they can generally lead to exploded certified bounds for IBP training as we will show in this paper.

### 2.3 Batch Normalization for DNN Training

Batch normalization (BN) (Ioffe & Szegedy, 2015) is originally proposed to improve DNN training by reducing interval covariate shift. More recently, Santurkar et al. (2018) instead suggests that BN actually improves DNN training by smoothing the loss landscape without the necessity of reducing internal covariate shift, and BN can accelerate DNN training (Van Laarhoven, 2017). In this paper, we identify the extra benefit of using BN in IBP training.

## 3 Methodology

### 3.1 Notations and Definitions

We focus on improving IBP training, and we consider a commonly adopted $\ell_\infty$ perturbation setting in adversarial robustness on a $K$-way classification task. For a DNN $f_\theta(\mathbf{x})$ with clean input $\mathbf{x}$, there can be some perturbation $\delta$ satisfying $\|\delta\|_\infty \leq \epsilon$, and the actual perturbed input to the model is $\mathbf{x} + \delta$.

In robustness verification for achieving certified robustness, we verify whether

$$[f_\theta(\mathbf{x} + \delta)]_y - [f_\theta(\mathbf{x} + \delta)]_i > 0, \quad \forall i, \delta \text{ s.t. } i \neq y, \|\delta\|_\infty \leq \epsilon, \tag{2}$$

holds true, where $[f_\theta(\mathbf{x}+\delta)]_i$ is the logits score for class $i$ and $y$ is the ground-truth. This is equivalent to verifying whether the DNN provably makes correct prediction for all input $\mathbf{x} + \delta$ ($\|\delta\|_\infty \leq \epsilon$). For network $f_\theta$, we assume that there are $m$ hidden affine layers (either convolutional or fully-connected layers) with ReLU activation. We use $\mathbf{h}_i$ to denote the pre-activation output value of the $i$-th layer, and $\mathbf{h}_{i,j}$ denotes the $j$-th neuron in the $i$-th layer. We also use $\mathbf{z}_i = \text{ReLU}(\mathbf{h}_i)$ to denote the post-activation value. For a convolutional or fully-connected layer, we use $\mathbf{W}_i$ and $\mathbf{b}_i$ to denote its parameters, where $\mathbf{W}_i \in \mathbb{R}^{r_i \times n_i}, \mathbf{b} \in \mathbb{R}^{r_i}$, and $r_i$ and $n_i$ are called the "fan-out" and "fan-in" number of the layer respectively (He et al., 2015b). This is straightforward for a fully-connected layer, and for a convolutional layer with kernel size $k$, $c_{\text{in}}$ input channels and $c_{\text{out}}$ output channels, we can still view the convolution as an affine transformation with $n_i = k^2 c_{\text{in}}$ and $r_i = c_{\text{out}}$. In particular, we use $\mathbf{h}_0 = \mathbf{x} + \delta$ to denote the input layer perturbed by $\delta$ ($\mathbf{z}_0$ is not applicable).

For IBP (Mirman et al., 2018; Gowal et al., 2018), it computes and propagates lower and upper bound intervals layer by layer until the last layer or the verification objective. For pre-activation $\mathbf{h}_i$, its interval bounds can be denoted as $[\underline{\mathbf{h}}_i, \overline{\mathbf{h}}_i]$, where $\underline{\mathbf{h}}_i \leq \mathbf{h}_i \leq \overline{\mathbf{h}}_i$ ($\forall \|\delta\|_\infty \leq \epsilon$). Similarly, there are also post-activation interval bounds $[\underline{\mathbf{z}}_i, \overline{\mathbf{z}}_i]$. Finally Eq. (2) can be verified by checking the lower bound of $[f_\theta(\mathbf{x} + \delta)]_y - [f_\theta(\mathbf{x} + \delta)]_i$.

## 3.2 Issues in Existing Certified Robust Training

In this section, we analyze the issues in existing IBP training. In particular, we identify two issues, including exploded bounds at initialization, and also the imbalance between ReLU activation states.

### 3.2.1 Exploded Bounds at Initialization

For simplicity, we assume the network has a feedforward architecture in this analysis, but the analysis can also be easily extended to other architectures. For affine layer $\mathbf{h}_i = \mathbf{W}_i \mathbf{z}_{i-1} + \mathbf{b}_i$, the IBP bound computation is as follows:

$$\underline{\mathbf{h}}_i = \mathbf{W}_{i,+}\underline{\mathbf{z}}_{i-1} + \mathbf{W}_{i,-}\overline{\mathbf{z}}_{i-1} + \mathbf{b}_i, \quad \overline{\mathbf{h}}_i = \mathbf{W}_{i,+}\overline{\mathbf{z}}_{i-1} + \mathbf{W}_{i,-}\underline{\mathbf{z}}_{i-1} + \mathbf{b}_i, \tag{3}$$

where $\mathbf{W}_{i,+}$ stands for retaining positive elements in $\mathbf{W}_i$ only while setting other elements to zero, and vice versa for $\mathbf{W}_{i,-}$. $\mathbf{h}_i$ can be viewed as a function with the post-activation value of the previous layer $\mathbf{z}_i$ as input, denoted as $\mathbf{h}_i(\mathbf{z}_i)$. In Eq. (3), the IBP bounds guarantee that $\underline{\mathbf{h}}_i \leq \mathbf{h}_i(\mathbf{z}_i) \leq \overline{\mathbf{h}}_i$ ($\forall \underline{\mathbf{z}}_i \leq \mathbf{z}_i \leq \overline{\mathbf{z}}_i$) for element-wise "$\leq$".

We then check the tightness of the interval bounds:

$$\Delta_i = \overline{\mathbf{h}}_i - \underline{\mathbf{h}}_i = |\mathbf{W}_i|(\overline{\mathbf{z}}_{i-1} - \underline{\mathbf{z}}_{i-1}) = |\mathbf{W}_i|\delta_{i-1}, \tag{4}$$

where $\Delta_i$ denotes the gap between the upper and lower bounds, which can reflect the tightness of the bounds, and $|\mathbf{W}_i|$ stands for taking the absolute value element-wise. At initialization, we assume that each $\mathbf{W}_i$ independently follows a distribution with zero mean and variance $\sigma_i^2$, and the distribution is symmetric about 0. For a vector or matrix with independent elements following the same distribution, we use $\mathbb{E}(\cdot)$ to denote the expectation of this distribution. We can view each element in vector $\Delta_i$ as a random variable that follows the same distribution, and we denote its expectation as $\mathbb{E}(\Delta_i)$, to measure the expected tightness at layer $i$. As $\mathbf{W}_i$ and $\delta_{i-1}$ are independent, we have $\mathbb{E}(\Delta_i) = n_i \mathbb{E}(|\mathbf{W}_i|)\mathbb{E}(\delta_{i-1})$. Detailed in Appendix D.1, we further have $\mathbb{E}(\delta_i) = \mathbb{E}(\text{ReLU}(\overline{\mathbf{h}}_i) - \text{ReLU}(\underline{\mathbf{h}}_i)) = \frac{1}{2}\mathbb{E}(\Delta_i)$, and

$$\mathbb{E}(\Delta_i) = \frac{n_i}{2}\mathbb{E}(|\mathbf{W}_i|)\mathbb{E}(\Delta_{i-1}). \tag{5}$$

Empirically, we can estimate $\mathbb{E}(\Delta_i)$ given a batch of concrete data, by taking the mean, and we use $\hat{\mathbb{E}}(\Delta_i)$ to denote the result of the empirical estimation.

We define a metric to characterize to what extent the certified bounds become looser, after propagating bounds from layer $i - 1$ to layer $i$:

**Definition 1.** *We define the difference gain when bounds are propagated from layer $i - 1$ to layer $i$:*

$$\mathbb{E}(\Delta_i)/\mathbb{E}(\Delta_{i-1}) = \frac{n_i}{2}\mathbb{E}(|\mathbf{W}_i|). \tag{6}$$

*Bounds are considered to be stable if the difference gain $\mathbb{E}(\Delta_i)/\mathbb{E}(\Delta_{i-1})$ is close to 1.*

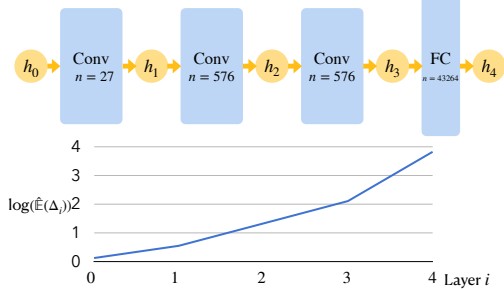

Figure 1: We show that certified bounds explode at initialization, in a simple untrained CNN (the classification layer is omitted) using Xavier initialization. We plot $\log \hat{\mathbb{E}}(\Delta_i)$ for each layer $i$.

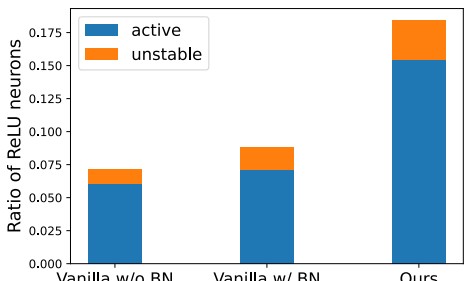

Figure 2: Ratios of active and unstable ReLU neurons for CNN-7 on CIFAR-10 with different settings. The vanilla ones are not regularized, and "vanilla (w/o BN)" does not use BN either.

A large difference gain indicates exploded bounds, but it cannot be much smaller than 1 either to avoid signal vanishing in the model. We find that weight initialization in prior works have large difference gain values especially for layers with larger $n_i$. For example, for the widely used Xavier initialization (Glorot & Bengio, 2010), the difference gain is $\frac{1}{4}\sqrt{n_i}$, and it can be as large as 45.25 when $n_i = 32768$ for a fully-connected layer in experiments. This indicates that certified bounds explode at initialization. We illustrate the bound explosion in Figure 1, and in Appendix A, we list the difference gain of each existing initialization method in Table 5. As a result, long warmup schedules are important in previous works, to gradually tighten certified bounds and ease training, but this is inefficient.

### 3.2.2 Imbalanced ReLU Activation States

We show another issue in existing certified training, where the models have a bias towards *inactive ReLU neurons*. Here "inactive ReLU neurons" are defined as neurons with non-positive pre-activation upper bounds ($\overline{\mathbf{h}}_{i,j} \leq 0$), i.e., they are always inactive regardless of input perturbations. Similarly, *active ReLU neurons* have non-negative pre-activation lower bounds ($\underline{\mathbf{h}}_{i,j} \geq 0$). There are also *unstable ReLU neurons* with uncertain activation states given different input perturbations ($\underline{\mathbf{h}}_{i,j} \leq 0 \leq \overline{\mathbf{h}}_{i,j}$). In IBP training, inactive neurons have tighter bounds than active and unstable ones as shown in Figure 5 in Appendix B, and thus the optimization tends to push the neurons to be inactive. We show this imbalance ReLU status in Figure 2 (vanilla w/o BN), and it is more severe when the warmup is shorter as shown in Appendix B.7. Too many inactive neurons indicates that many neurons are essentially unused or dead, which will harm the model's capacity and block gradients as discussed by Lu et al. (2019) on standard training.

### 3.3 The Proposed Method

To address the aforementioned issues, we propose our method in three parts: 1) We derive a new weight initialization for IBP training to stabilize the tightness of bounds at initialization; 2) We propose to fully add BN to mitigate the ReLU imbalance and stabilize the variance of bounds, while models in prior works did not have BN for some or all the layers. 3) We further propose regularizations to explicitly stabilize the tightness and the balance of ReLU states during warmup.

### 3.3.1 IBP initialization

We propose a new *IBP initialization* for IBP training. Specifically, we independently initialize each element in $\mathbf{W}_i$ following a normal distribution $\mathcal{N}(0, \sigma_i^2)$, and we aim to choose a value for $\sigma_i$ such that the *difference gain* defined in Eq. (6) is exactly 1. When elements in $\mathbf{W}_i$ follow the normal distribution, we have $\mathbb{E}(|\mathbf{W}_i|) = \sqrt{2/\pi}\sigma_i$, and thereby we take $\sigma_i = \frac{\sqrt{2\pi}}{n_i}$, which makes the difference gain $\frac{n_i}{2}\mathbb{E}(|\mathbf{W}_i|)$ exactly 1. This initialization can further be calibrated for non-feedforward networks such as ResNet as we discuss in Appendix A.3.

### 3.3.2 Batch Normalization

Batch normalization (BN) (Ioffe & Szegedy, 2015) normalizes the input of each layer to a distribution with stable mean and variance. It can improve the optimization for DNN as shown in prior works for standard DNN training (Ioffe & Szegedy, 2015; Van Laarhoven, 2017; Santurkar et al., 2018). In addition, for IBP training, BN can normalize the variance of bounds, and it can also improve the balance of ReLU activation states by shifting the center of upper and lower bounds to zero (before the additional linear transformation which comes after the normalization). In prior certified training works (Gowal et al., 2018; Zhang et al., 2020; Xu et al., 2020), they only used BN for some layers in some models but not all layers, and they did not identify the benefit of BN in certified training. We empirically demonstrate that fully adding BN to each affine layer can significantly mitigate the imbalance ReLU issue and improve IBP training. We follow the BN implementation by Wong et al. (2018); Xu et al. (2020) for certified training, where the shifting and scaling parameters are computed from unperturbed data.

Note that our previous analysis on IBP initialization considers a network without BN. BN which rescales the output of each layer can still affect the tightness of IBP bounds, and the effect of IBP initialization may be weakened. This is a limitation of the proposed initialization which could possibly be improved by considering the effect of BN in future work. Nevertheless, in Appendix A.4, we empirically show that BN still does not cancel out the effect of IBP initialization.

### 3.3.3 Warmup Regularization

To further address the aforementioned two issues in Sec. 3.2, and to explicitly stabilize the tightness of certified bounds and balance ReLU neuron states, we add two regularizers in the warmup stage of IBP training, The regularizers are principled and motivated by the two issues we discover.

**Bound tightness regularizer**   Similar to the goal of stabilizing certified bounds at initialization, we also expect to keep the mean value of $\Delta_i$ in the current batch, $\hat{\mathbb{E}}(\Delta_i)$, stable along the warmup. Note that $\hat{\mathbb{E}}(\Delta_i)$ is empirically computed from a concrete batch and different from the expectation $\mathbb{E}(\Delta_i)$ at initialization In the initialization, we aim to make $\mathbb{E}(\Delta_i) \approx \mathbb{E}(\Delta_{i-1})$. Here, we relax the goal to making $\tau\hat{\mathbb{E}}(\Delta_i) \leq \hat{\mathbb{E}}(\Delta_0)$ with a configurable tolerance value $\tau$ $(0 < \tau \leq 1)$, to balance the regularization power and the model capacity. We add the following regularization term:

$$\mathcal{L}_{\text{tightness}} = \frac{1}{\tau m} \sum_{i=1}^{m} \text{ReLU}(\tau - \frac{\hat{\mathbb{E}}(\Delta_0)}{\hat{\mathbb{E}}(\Delta_i)}), \tag{7}$$

where the training is penalized only when $\tau\hat{\mathbb{E}}(\Delta_i) > \hat{\mathbb{E}}(\Delta_0)$ due to the clipping effect by $\text{ReLU}(\cdot)$.

**ReLU activation states balancing regularizer**   To balance ReLU activation states, we expect to balance the impact of active ReLU neurons and inactive neurons respectively. Here, we consider the center of the interval bound, $\mathbf{c}_i = (\underline{\mathbf{h}}_i + \overline{\mathbf{h}}_i)/2$, and we model the impact as the contribution of each type of neurons to the mean and variance of the whole layer, i.e., $\hat{\mathbb{E}}(\mathbf{c}_i)$ and $\text{Var}(\mathbf{c}_i)$ respectively. Note that in the beginning almost all neurons are unstable, and gradually most neurons become either active or inactive. Therefore, we add this regularizer only when there is at least one active neuron and one inactive neuron, which generally holds true unless at the training start. We use $\alpha_i$ to denote the ratio between the contribution of the active neurons and inactive neurons respectively to $\hat{\mathbb{E}}(\mathbf{c}_i)$, and similarly we use $\beta_i$ to denote the ratio of contribution to $\text{Var}(\mathbf{c}_i)$. They are computed as:

$$\alpha_i = \frac{\sum_j \mathbb{I}(\underline{\mathbf{h}}_{i,j} > 0)\mathbf{c}_{i,j}}{-\sum_j \mathbb{I}(\overline{\mathbf{h}}_{i,j} < 0)\mathbf{c}_{i,j}}, \qquad \beta_i = \frac{\sum_j \mathbb{I}(\underline{\mathbf{h}}_{i,j} > 0)(\mathbf{c}_{i,j} - \hat{\mathbb{E}}(\mathbf{c}_i))^2}{\sum_j \mathbb{I}(\overline{\mathbf{h}}_{i,j} < 0)(\mathbf{c}_{i,j} - \hat{\mathbb{E}}(\mathbf{c}_i))^2},$$

and in general $\alpha_i, \beta_i > 0$. We regard that the activation states are roughly balanced if $\alpha_i$ and $\beta_i$ are close to 1. With the same aforementioned tolerance $\tau$, we expect to make $\tau \leq \alpha_i, \beta_i \leq 1/\tau$, which is equivalent to making $\min(\alpha_i, 1/\alpha_i) \geq \tau$, $\min(\beta_i, 1/\beta_i) \geq \tau$. Thereby we design the following regularization term:

$$\mathcal{L}_{\text{relu}} = \frac{1}{\tau m} \sum_{i=1}^{m} \left( \text{ReLU}(\tau - \min(\alpha_i, \frac{1}{\alpha_i})) + \text{ReLU}(\tau - \min(\beta_i, \frac{1}{\beta_i})) \right). \tag{8}$$

## 3.4 Training Objectives

Certified robust training solves the robust optimization problem as Eq. (1), and when the inner maximization is verifiably solved, the base training objective without regularization is:

$$\mathcal{L}_{\text{rob}} = \overline{L}(f_\theta, \mathbf{x}, y, \epsilon), \quad \text{where } \overline{L}(f_\theta, \mathbf{x}, y, \epsilon) \geq \max_{\|\delta\|_\infty \leq \epsilon} L(f_\theta(\mathbf{x} + \delta), y), \tag{9}$$

such that $\overline{L}(f_\theta, \mathbf{x}, y, \epsilon)$ is an upper bound of $L(f_\theta(\mathbf{x} + \delta), y)$ given by a robustness verifier, e.g., IBP. In our proposed method, we first initialize the parameters with our IBP initialization, and then we perform a *short* warmup with gradually increasing $\epsilon$ ($0 \leq \epsilon \leq \epsilon_{\text{target}}$), where $\epsilon_{\text{target}}$ stands for the target perturbation radius that is usually equal to or slightly larger than the maximum perturbation radius used for test. Our training objective $\mathcal{L}$ combines the ordinary objective Eq. (9) and the proposed regularizers:

$$\mathcal{L} = \mathcal{L}_{\text{rob}} + \lambda(\mathcal{L}_{\text{tightness}} + \mathcal{L}_{\text{relu}}), \tag{10}$$

where $\lambda$ is for balancing the regularizers and the original $\mathcal{L}_{\text{rob}}$ loss. For simplicity and efficiency, we use IBP to compute the bounds in $\mathcal{L}_{\text{rob}}$ and the regularizers. During warmup, we also gradually decrease $\lambda$ from $\lambda_0$ to 0 as $\epsilon$ grows, where $\lambda = \lambda_0(1 - \epsilon/\epsilon_{\text{target}})$. After warmup, we only use $\mathcal{L} = \mathcal{L}_{\text{rob}}$ for final training with $\epsilon_{\text{target}}$. Note that in the regularizers, the value of each ReLU$(\cdot)$ term has the same range $[0, \tau]$, and thus in Eq. (10) we directly sum up them without weighing them for simplicity. In test, we still only use pure IBP bounds without any other tighter method.

# 4 Experiments

In the experiments, we demonstrate the effectiveness of our proposed method for training certifiably robust neural networks more efficiently while achieving better or comparable verified errors.

## 4.1 Settings

We adopt three datasets, MNIST (LeCun et al., 2010), CIFAR-10 (Krizhevsky et al., 2009) and TinyImageNet (Le & Yang, 2015). Following Xu et al. (2020), we consider three model architectures: a 7-layer feedforward convolutional network (CNN-7), Wide-ResNet (Zagoruyko & Komodakis, 2016) and ResNeXt (Xie et al., 2017). According our discussion in Sec. 3.3.2, we also modify the models to fully add a BN after every convolutional or fully-connected layer. For target perturbation radii, we mainly use $\epsilon_{\text{target}} = 0.4$ for MNIST, $\epsilon_{\text{target}} = 8/255$ for CIFAR-10, and $\epsilon_{\text{target}} = 1/255$ for TinyImageNet, following prior works, and we provide results on other perturbation radii in Appendix B.3. We provide more implementation details in Appendix C. We mainly compare with the following SOTA baselines on all the settings (note that in our main results, we also make these baselines use models with full BNs unless otherwise indicated):

- Vanilla IBP (Gowal et al., 2018) with existing initialization and no warmup regularizer. We use the default Xavier initialization in PyTorch, and we find that orthogonal initialization originally used by Gowal et al. (2018) does not improve the performance here.

- CROWN-IBP (Zhang et al., 2020) with linear relaxation bounds by CROWN (Zhang et al., 2018) during warmup. We use the generalized and accelerated version with loss fusion by Xu et al. (2020), while the original version is $O(K)$ (the number of classes) more costly. During the warmup, it combines bounds by IBP and linear relaxation with weight $\epsilon/\epsilon_{\text{target}}$ and $(1 - \epsilon/\epsilon_{\text{target}})$ respectively.

## 4.2 Certified Robust Training with Short Warmup

We conduct certified robust training using relatively short warmup schedules to demonstrate the effectiveness of our proposed techniques for fast training. We show the results in Table 1 for MNIST, CIFAR-10 and Table 2 for TinyImageNet. Compared to Vanilla IBP and CROWN-IBP, our improved IBP training consistently achieves lower standard errors and verified errors under same schedules respectively, where BN is added to the models for all these three training methods. We find that CROWN-IBP with loss fusion (Xu et al., 2020) tends to require a larger number of epochs to obtain good results and it sometimes underperform Vanilla IBP under short schedules, but disabling loss fusion can make it much more costly and unscalable. In terms of the best results, we achieve verified error 10.82% on MNIST $\epsilon_{\text{target}} = 0.4$, 65.03% on CIFAR-10 $\epsilon_{\text{target}} = 8/255$, and 82.36% on

Table 1: Standard and verified error rates (%) of models trained with different methods respectively on MNIST ($\epsilon_{\text{target}} = 0.4$) and CIFAR-10 ($\epsilon_{\text{target}} = 8/255$). Schedule is represented as the total number of epochs and the number of epochs in each of the three phases with $\epsilon = 0$, increasing $\epsilon \in (0, \epsilon_{\text{target}})$ and final $\epsilon = \epsilon_{\text{target}}$ respectively. We report the mean and standard deviation of the results on 5 repeats for CNN-7 and 3 repeats for Wide-ResNet and ResNeXt respectively. All models include BN after every layer (see Sec. 3.3.2). We also report the best run in "Ours (best)" since main results in prior works did not have repeats. Literature results with the "†" mark are concurrent works.

| Dataset | Schedule (epochs) | Method | CNN-7 (with full BN) Standard | Verified | Wide-ResNet (with full BN) Standard | Verified | ResNeXt (with full BN) Standard | Verified |
|---|---|---|---|---|---|---|---|---|
| MNIST | 70 (0+20+50) | Vanilla IBP | $2.59 \pm 0.06$ | $12.03 \pm 0.09$ | $3.18 \pm 0.05$ | $12.93 \pm 0.17$ | $4.09 \pm 0.46$ | $15.36 \pm 0.94$ |
| | | CROWN-IBP [a] | $2.75 \pm 0.12$ | $12.04 \pm 0.22$ | $3.39 \pm 0.05$ | $13.10 \pm 0.15$ | $4.22 \pm 0.53$ | $15.24 \pm 0.78$ |
| | | Ours | $\mathbf{2.33 \pm 0.08}$ | $\mathbf{11.03 \pm 0.13}$ | $\mathbf{2.77 \pm 0.02}$ | $\mathbf{11.76 \pm 0.07}$ | $\mathbf{3.22 \pm 0.08}$ | $\mathbf{13.43 \pm 0.17}$ |
| | | Ours (best) | **2.20** | **10.82** | 2.75 | 11.69 | 3.17 | 13.20 |
| | | Literature results | Warmup | | | Total (epochs) | Standard | Verified |
| | | Gowal et al. (2018) | (2K+10K) steps | | | 100 | 1.66 | 15.01 [b] |
| | | Zhang et al. (2020) | (9 + 51) epochs | | | 200 | 2.17 | 12.06 |
| | | [†]IBP+ParamRamp (Lyu et al., 2021) [e] | (9 + 51) epochs | | | 200 | 2.16 | 10.88 |
| | | [†]CROWN-IBP+ParamRamp (Lyu et al., 2021) [e] | (9 + 51) epochs | | | 200 | 2.36 | 10.61 |
| CIFAR-10 | 70 (1+20+49) | Vanilla IBP | $58.72 \pm 0.27$ | $69.88 \pm 0.10$ | $58.85 \pm 0.22$ | $69.77 \pm 0.32$ | $60.10 \pm 0.27$ | $71.19 \pm 0.21$ |
| | | CROWN-IBP [a] | $63.19 \pm 0.36$ | $71.29 \pm 0.19$ | $62.76 \pm 0.23$ | $71.82 \pm 0.30$ | $64.75 \pm 0.50$ | $72.50 \pm 0.20$ |
| | | Ours | $\mathbf{56.64 \pm 0.48}$ | $\mathbf{68.81 \pm 0.24}$ | $\mathbf{56.74 \pm 0.40}$ | $\mathbf{68.71 \pm 0.29}$ | $\mathbf{59.33 \pm 0.86}$ | $\mathbf{70.62 \pm 0.59}$ |
| | 160 (1+80+79) | Vanilla IBP | $53.80 \pm 0.71$ | $67.01 \pm 0.29$ | $54.31 \pm 0.46$ | $67.45 \pm 0.21$ | $55.23 \pm 0.12$ | $68.28 \pm 0.15$ |
| | | CROWN-IBP [a] | $58.76 \pm 0.76$ | $69.67 \pm 0.38$ | $60.39 \pm 0.33$ | $70.07 \pm 0.42$ | $61.08 \pm 0.35$ | $71.26 \pm 0.11$ |
| | | Ours | $\mathbf{51.72 \pm 0.40}$ | $\mathbf{65.58 \pm 0.32}$ | $\mathbf{51.95 \pm 0.27}$ | $\mathbf{65.91 \pm 0.14}$ | $\mathbf{53.68 \pm 0.33}$ | $\mathbf{66.91 \pm 0.40}$ |
| | | Ours (best) | **51.06** | **65.03** | 51.63 | 65.72 | 53.38 | 66.41 |
| | | Literature results | Warmup | | | Total (epochs) | Standard | Verified |
| | | Gowal et al. (2018) | (5K+50K) steps | | | 3,200 | 50.51 | 68.44 [c] |
| | | Zhang et al. (2020) | (320 + 1600) epochs | | | 3,200 | 54.02 | 66.94 |
| | | Balunovic & Vechev (2020) | N/A [d] | | | 800 | 48.3 | 72.5 |
| | | Xu et al. (2020) | (100 + 800) epochs | | | 2,000 | 53.71 | 66.62 |
| | | [†]IBP+ParamRamp (Lyu et al., 2021) [e] | (320 + 1600) epochs | | | 3,200 | 55.28 | 67.09 |
| | | [†]CROWN-IBP+ParamRamp (Lyu et al., 2021) [e] | (320 + 1600) epochs | | | 3,200 | 51.94 | 65.08 |
| | | [†]$\ell_\infty$-dist net (other architecture) (Zhang et al., 2021) [f] | N/A [f] | | | 800 | 48.32 | 64.90 |

[a] CROWN-IBP here follows Xu et al. (2020) with loss fusion for efficiency, but we found it does not perform well with a short training schedule under our settings and usually requires a longer schedule to achieve good results.

[b] Some test results in Gowal et al. (2018) are obtained with costly mixed integer programming (MIP) and linear programming (LP); we take IBP verified errors for fair comparison following Zhang et al. (2020).

[c] Additional PGD adversarial training was involved for this result, according to Zhang et al. (2020).

[d] Balunovic & Vechev (2020) used a different training scheme and train the network layer by layer.

[e] Lyu et al. (2021) use IBP-based and CROWN-IBP-based training respectively with their parameterized activation, and they use a tighter linear bound propagation method for testing instead of IBP.

[f] Zhang et al. (2021) use a very different model architecture with $\ell_\infty$ distance neurons rather than traditional DNNs, but still need a long schedule on both $\epsilon$ and $\ell_p$ norm where $p$ is gradually increased until $\infty$.

Table 2: Standard and verified error rates (%) on TinyImageNet ($\epsilon_t = 1/255$). The best result in literature (Xu et al., 2020) has a standard error of 72.18% and verified error of 84.14% using 800 epochs. We achieve 82.36% verified error using only 80 epochs.

| Model (with full BN) | Schedule (epochs) | Vanilla IBP Standard | Verified | CROWN-IBP Standard | Verified | Ours Standard | Verified |
|---|---|---|---|---|---|---|---|
| CNN-7 | 80 (1+10+69) | 75.50 | 82.92 | 76.00 | 82.81 | 75.20 | **82.45** |
| | 80 (1+20+59) | 74.68 | 82.84 | 76.27 | 83.35 | 74.29 | **82.36** |
| Wide-ResNet [a] | 80 (1+10+69) | 75.89 | 83.00 | 75.85 | 83.65 | 74.90 | **82.49** |
| | 80 (1+20+59) | 75.65 | 83.17 | 75.95 | 83.08 | 74.59 | **82.75** |
| ResNeXt | 80(1+10+69) | 82.39 | 87.15 | 85.47 | 89.11 | 80.20 | **85.77** |
| | 80 (1+20+59) | 81.72 | 87.10 | 80.81 | 86.43 | 78.91 | **85.78** |

[a] The Wide-ResNet model used here is 5 times smaller than the one used in Xu et al. (2020) to save training time. Additionally, we include BN after every layer in all models (see Section 3.3.2).

TinyImageNet $\epsilon_{\text{target}} = 1/255$, which makes a notable improvement over literature SOTA (Gowal et al., 2018; Xu et al., 2020) that used long training schedules. Compared to concurrent works (Lyu et al., 2021; Zhang et al., 2021) which use different improvement techniques, we have comparable verified errors, but they still need long training schedules. For reference, we tried Zhang et al. (2021) which used a different architecture with "$\ell_\infty$ distance neurons" rather than convolution-based DNNs. On CIFAR-10 using 160 total epochs by reducing their training schedule proportionally, their verified error is 68.44% which is much higher than ours. Overall, the results demonstrate that our improved IBP training is effective for more efficient certified robust training with a shorter warmup.

### 4.3 Comparison on Training Cost

Table 3: Comparison of estimated time cost (seconds), for CNN-7 on CIFAR-10. We report the total time, and also the per-epoch time during three training phases of $\epsilon$ schedule for methods with a short warmup. Literature results with the "†" mark are considered as concurrent.

| | Method | Epochs | Epoch time in each phase (s) | | | Total time (s) |
|---|---|---|---|---|---|---|
| | | | 0 | $(0, \epsilon_{\text{target}})$ | $\epsilon_{\text{target}}$ | |
| Literature Results | IBP (Gowal et al., 2018) | 3200 | | | | $40496 \times 4$ [a] |
| | CROWN-IBP (w/o loss fusion) (Zhang et al., 2020) | 3200 | | - | | $91288 \times 4$ [a] |
| | CROWN-IBP (Xu et al., 2020) | 2000 | | | | $52362 \times 4$ [a] |
| | †IBP+ParamRamp (Lyu et al., 2021) | 3200 | | | | $40496 \times 4 \times 1.09$ [b] |
| | †CROWN-IBP+ParamRamp (Lyu et al., 2021) | 3200 | | - | | $91288 \times 4 \times 1.51$ [b] |
| Short Warmup | Vanilla IBP | 160 | 30.0 | 54.8 | 54.8 | 8747.9 |
| | CROWN-IBP | 160 | 30.0 | 78.5 | 54.8 | 10641.3 |
| | Ours | 160 | 64.0 | 64.0 | 54.8 | 9512.3 |

[a] 4 GPUs were used and their models are slightly different (we add BN after every layer).
[b] The factors 1.09 and 1.51 are the overhead of their method reported by (Lyu et al., 2021) when combining with IBP or CROWN-IBP.

We compare the training cost using a single NVIDIA RTX 2080 Ti GPU. For methods using short warmup, we measure the per-epoch time cost during three different phases, namely $\epsilon = 0$, $0 < \epsilon < \epsilon_{\text{target}}$, and $\epsilon = \epsilon_{\text{target}}$, and we then estimate the total training time according to the schedule. We use gradient accumulation wherever needed to fit the training into the memory of a single GPU. We also compare with total time cost with literature methods using long schedules. We show the results of CNN-7 for CIFAR-10 in Table 3, and other settings in Appendix B.1. For $\epsilon = 0$, Vanilla IBP and CROWN-IBP use regular training while we compute IBP bounds for regularization and have a small overhead, but this phase is extremely short (no more than 1 epoch here). For $0 < \epsilon < \epsilon_{\text{target}}$, our method has a small overhead on regularizers compared to Vanilla IBP, while CROWN-IBP using linear relaxation can be more costly. For $\epsilon = \epsilon_{\text{target}}$, all the three methods use the same pure IBP.

For total time on CIFAR-10 with the same 160-epoch schedule, we only have a small overhead of around $9\% \sim 13\%$ compared to Vanilla IBP and the cost is still around $12\% \sim 23\%$ lower than CROWN-IBP, while we achieve lower verified errors than the baselines under such short warmup schedules (see Table 1). And importantly, compared to literature using long training schedules, we significantly reduce the number of training epochs and the total training time (e.g., Xu et al. (2020) is around $20\times$ more costly than ours in total).

### 4.4 Ablation Study and Discussions

In this section, we empirically verify whether each part of our modification contributes to the improvement and whether they behave as we expect. We conduct an ablation study and also plot the curve of the regularization terms to reflect the bound tightness and ReLU balance during training.

We use CIFAR-10 with the currently best CNN-7 model under the "$1 + 20$" and "$1 + 80$" warmup schedules as used in Table 1. We report the results in Table 4. The first three rows show that fully adding BN improves the training when vanilla IBP is used, and it is important to add BN for the fully-connected layer, which was missed in prior works. Based on the improved model structure, adding both IBP initialization and warmup regularization further improves the performance, and removing either of these parts leads to a degraded performance.

We notice that adding IBP initialization without warmup regularization may not improve the verified error. A factor is that IBP initialization can reduce the variance of the outputs (see Appendix D.2), and it may harm the training during the early warmup, when $\epsilon$ is small and certified training is close to standard training. Also, the effect of initialization can be weakened when $\epsilon$ is much smaller than $\epsilon_{\text{target}}$. But the warmup regularization can continue to tighten the bounds, and the IBP initialization can benefit the optimization for the tightness regularizer. Nevertheless, IBP initialization is more beneficial for deep models where the exploded bound issue is more severe (see Appendix B.8).

It is also important to fully add BN to make the warmup regularization work well. BN can normalize the variance of the layers, so the tightness regularizer can more effectively tighten certified bounds w.r.t. the stable variance; otherwise the the training may trivially optimize tightness regularizer by making the magnitude of the network output small.

Table 4: Standard error and verified error rates (%) in the ablation study with CNN-7 on CIFAR-10. "BN-Conv" stands for BN after each convolutions, and "BN-FC" stands for BN after the hidden fully-connected layer. "✓" means that the component is enabled, and "×" means that the component is disabled. We repeat each setting for 5 times and report the mean and standard deviation.

| BN-Conv | BN-FC | IBP Initialization | $\mathcal{L}_{\text{tightness}}$ | $\mathcal{L}_{\text{relu}}$ | 70 (1+20+49) epochs | | 160 (1+80+79) epochs | |
|---|---|---|---|---|---|---|---|---|
| | | | | | Standard | Verified | Standard | Verified |
| × | × | × | × | × | 59.33±0.70 | 70.18±0.18 | 57.08±0.29 | 69.43±0.28 |
| ✓ | × | × | × | × | 61.95±0.80 | 71.12±0.42 | 57.21±0.65 | 69.21±0.30 |
| ✓ | ✓ | × | × | × | 58.72±0.27 | 69.88±0.10 | 53.80±0.71 | 67.01±0.29 |
| ✓ | ✓ | ✓ | × | × | 58.93±0.29 | 69.60±0.35 | 54.59±0.64 | 67.63±0.34 |
| ✓ | ✓ | ✓ | ✓ | × | 56.76±0.38 | 68.96±0.49 | 53.08±0.26 | 66.74±0.20 |
| ✓ | ✓ | ✓ | × | ✓ | 58.49±0.42 | 69.38±0.23 | 53.29±0.76 | 66.46±0.44 |
| ✓ | ✓ | × | ✓ | ✓ | 58.79±0.40 | 69.29±0.28 | 52.45±0.34 | 66.34±0.38 |
| ✓ | ✓ | ✓ | ✓ | ✓ | **56.64±0.48** | **68.81±0.24** | **51.72±0.40** | **65.58±0.32** |

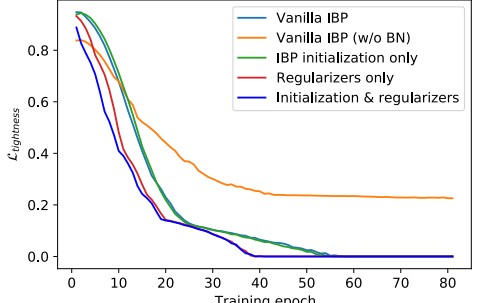

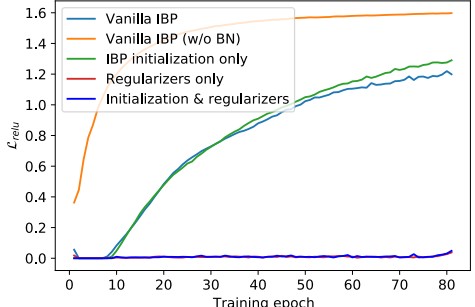

Figure 3: $\mathcal{L}_{\text{tightness}}$ during warmup. $\mathcal{L}_{\text{tightness}}$ is optimized only for "regularizers only" and "initialization & regularizers" setting, and BN is fully added to every layer except for "Vanilla IBP (w/o BN)".

Figure 4: $\mathcal{L}_{\text{relu}}$ during warmup, under same setting as in Figure 3.

Finally, we also plot the training curves of the regularizers to confirm if the regularizers are effectively optimized, so that the bound tightness and ReLU balance are indeed improved. Note that for the settings without regularizers, we only plot but not optimize the regularizers. In Figure 3, we plot $\mathcal{L}_{\text{tightness}}$. By using the regularization in training, $\mathcal{L}_{\text{tightness}}$ descends faster, and further adding the IBP initialization leads to even faster descent during the early epochs. In Figure 4, we show that the $\mathcal{L}_{\text{relu}}$ is indeed under control when we optimize it, while $\mathcal{L}_{\text{relu}}$ could gradually grow larger when the it is not added in training. Notably, when BN is removed and the regularization term is not optimized (Vanilla IBP (w/o BN)), $\mathcal{L}_{\text{relu}}$ becomes extremely large in later epochs, and $\mathcal{L}_{\text{tightness}}$ is also large in the end, which suggests that the training is hampered.

## 5   Conclusion

In this paper, we identify two issues in existing certified robust training methods regarding exploded bounds and imbalanced ReLU neuron states. To address these issues based on IBP training, we propose an IBP initialization and warmup regularization, and we also identify the benefit of fully adding BN. With our improvements, we demonstrate that we are able to achieve better verified errors using much shorter warmup and training schedules compared to literatures under the same convolution-based network architecture, for fast certified robust training.

## Funding Disclosure

This work is supported in part by NSF under IIS-1901527, IIS-2008173, IIS-2048280 and by Army Research Laboratory under agreement number W911NF-20-2-0158.

Additional revenues related to this work: Zhouxing Shi and Yihan Wang were interns at JD AI Research when this work was partly done; Cho-Jui Hsieh has part-time employment at Amazon.

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
