# A  Supplementary Illustrations for Motivation and Methodology

## A.1  List of Initialization Methods in Prior Works

Table 5: List of several weight initialization methods and their *difference gain*. We show each difference gain in both closed form, and also empirical values when $n_i \in \{27, 576, 1152, 32768\}$ for a 7-layer CNN model (without BN). The concrete values are obtained by computing the mean of 100 random trials respectively. For orthogonal initialization, obtaining a closed form of difference gain is non-trivial so we omit its closed-form result, but it has large difference gains under empirical measurements.

| Method | Adopted by | Difference Gain Closed form | $n_i = 27$ | $n_i = 576$ | $n_i = 1152$ | $n_i = 32768$ |
|---|---|---|---|---|---|---|
| Xavier (uniform) (Glorot & Bengio, 2010) | Zhang et al. (2020); Xu et al. (2020) | $\frac{1}{4}\sqrt{n_i}$ | 1.30 | 6.00 | 8.48 | 45.25 |
| Xavier (Gaussian) (Glorot & Bengio, 2010) | - | $\sqrt{\frac{1}{2\pi}}\sqrt{n_i}$ | 2.07 | 9.57 | 13.54 | 72.2 |
| Kaiming (uniform) (He et al., 2015b) | - | $\frac{\sqrt{3}}{4}\sqrt{n_i}$ | 3.20 | 14.70 | 20.77 | 110.85 |
| Kaiming (Gaussian) (He et al., 2015b) | - | $\sqrt{\frac{1}{\pi}}\sqrt{n_i}$ | 2.93 | 13.54 | 19.15 | 102.13 |
| Orthogonal (Saxe et al., 2013) | Gowal et al. (2018) | - | 2.09 | 9.58 | 13.54 | 72.22 |
| IBP Initialization | This work | 1 | 1.01 | 1.00 | 1.00 | 1.00 |

In Table 5, we list several weight initialization methods and their corresponding difference gain (see Def. 1). Prior weight initialization methods lead to large difference gain values especially when $n_i$ is larger, which indicates exploded certified bounds at initialization. In contrast, our initialization yields a constant difference gain of 1 regardless of $n_i$.

While He et al. (2015b) proposed Kaiming initialization to stabilize the variance of each layer in standard DNN training, compared to Xavier initialization, it has even larger difference gain values and thus it tends to worsen the tightness of certified bounds here. For CNN-7 on CIFAR-10 using 160 total training epochs, if we make the Vanilla IBP baseline use Kaiming initialization, the verified error is $(68.07 \pm 0.30)\%$, which is worse than the baseline result using Xavier initialization, i.e., $(67.01 \pm 0.29)\%$. The empirical result aligns with our theoretical insight since Kaiming initialization has larger difference gain values.

## A.2  Illustration of IBP Relaxations for Different Neuron States

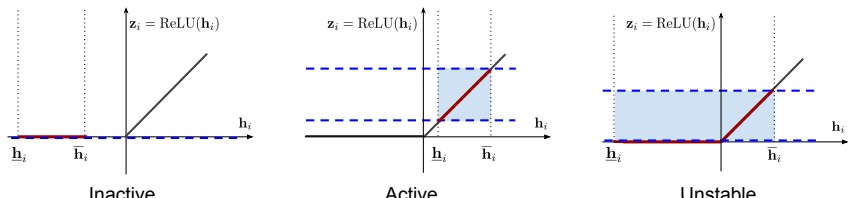

Figure 5: Three activation states of ReLU neurons determined by pre-activation lower and upper bounds and their corresponding IBP relaxations. The relaxed areas are shown in light blue.

In Figure 5, we illustrate IBP relaxations for ReLU neurons with the three different states respectively. Inactive neurons have no relaxation error compared with the other two kinds of neurons, and thus IBP training tends to prefer inactive neurons more to tighten certified bounds, compared to the other two ReLU neuron states. This leads to an imbalance in ReLU neuron states for vanilla IBP on models without BN. In this paper, we identify the benefit of fully adding BN layers to mitigate the imbalance, because BN normalizes pre-activation values. We also add a regularization to further encourage ReLU balance.

## A.3  IBP Initialization for Non-feedforward Networks

Our analysis in Section 3.2.1 is based on feedforward networks but it can also be easily extended to other architectures. On the weight initialization for standard DNN training, Hanin & Rolnick (2018); Arpit et al. (2019) extended the weight initialization to ResNet, which aimed to keep the variance stable. In IBP initialization, we want to make $\mathbb{E}(\Delta_i)$ stable instead, and we give an example

on ResNet. We consider a residual connection $\tilde{\mathbf{h}}_i = \mathbf{h}_i + \mathbf{h}_{i-1}$, and we want to make its tightness $\overline{\mathbf{h}}_i + \overline{\mathbf{h}}_{i-1} - (\underline{\mathbf{h}}_i + \underline{\mathbf{h}}_{i-1})$ stable, which equals to $\Delta_i + \Delta_{i-1}$. Our IBP initialization in Section 3.3.1 makes $\mathbb{E}(\Delta_i) \approx \mathbb{E}(\Delta_{i-1})$, and thereby $\mathbb{E}(\Delta_i + \Delta_{i-1}) \approx 2\mathbb{E}(\Delta_{i-1})$. Here we get an additional growth factor of 2, when propagating bounds from layer $i-1$ to layer $i$. This factor is a constant and does not depend on the fan-in number $n_i$. We can further remove this factor, we can divide the weight after each residual connection by 2 (this is equivalent to dividing $\tilde{\mathbf{h}}_i$ by 2 when it is used by subsequent layers).

### A.4 Effect of Batch Normalization on IBP Bound Tightness

Our analysis in Section 3.2.1 does not consider BN. In this section, we analyze the tightness of certified bounds when BN presents. As mentioned in Section 3.3.2, we use mean and variance estimation computed from clean data in BN (which is also the standard way). For the output bounds $\underline{\mathbf{h}}_i$ and $\overline{\mathbf{h}}_i$, we use $\underline{\mathbf{h}}'_i$ and $\overline{\mathbf{h}}'_i$ to denote the output bounds after BN. We have $\underline{\mathbf{h}}'_i = a_i \frac{\mathbf{h}_i - \mu(\mathbf{h}_i)}{\sigma(\mathbf{h}_i)} + b_i$, where $\mu(\mathbf{h}_i)$ and $\sigma(\mathbf{h}_i)$ stand for the estimated the mean and standard deviation respectively from clean output $\mathbf{h}_i$, and $a_i$ and $b_i$ are the weight and bias of BN. Similarly we can get $\overline{\mathbf{h}}'_i$. Therefore, to conduct a analysis similar to Sec. 3.2.1 for BN, we first need to estimate $\mu(\mathbf{h}_i)$ and $\sigma(\mathbf{h}_i)$, and then we can estimate $\overline{\mathbf{h}}'_i, \underline{\mathbf{h}}'_i$. Finally, we have $\Delta'_i = \overline{\mathbf{h}}'_i - \underline{\mathbf{h}}'_i$ to denote the bound tightness after BN.

At initialization, we assume elements in $\mathbf{h}_i$ are independently initialized following a zero-mean Gaussian distribution, and $\Delta'_i$ can be computed from the variance of the Gaussian distribution. However, after a single step of training, elements in $\mathbf{h}_i$ are no longer independent, and the mean and variance in BN are difficult to calculate explicitly. But we can empirically estimate them. Although when $\sigma(\mathbf{h}_i) < 1$ (which is true if we use IBP initialization to tighten certified bounds), $\Delta'_i$ will get larger than $\Delta_i$, i.e., bounds become looser after they are propagated through BN, we can show empirically that IBP initialization is still able to tighten the bounds in this situation.

In Table 6, we compare $\log(\hat{\mathbb{E}}(\Delta_m)/\hat{\mathbb{E}}(\Delta_0))$ of CNN-7 model with full BN on CIFAR-10 during the early epochs, where $m$ is the last layer of the model, with and without IBP initialization respectively. A smaller value indicates that the bounds are tighter. And we can see that the model with IBP initialization has smaller $\hat{\mathbb{E}}(\Delta_m)/\hat{\mathbb{E}}(\Delta_0)$ along these epochs and thus has tighter bounds.

| IBP Initialization | Epoch 1 | Epoch 2 | Epoch 3 | Epoch 4 |
|:---:|:---:|:---:|:---:|:---:|
| No | 16.29 | 15.21 | 13.08 | 11.90 |
| Yes | 11.56 | 12.42 | 11.97 | 11.24 |

Table 6: $\log(\hat{\mathbb{E}}(\Delta_m)/\hat{\mathbb{E}}(\Delta_0))$ at the first 5 epochs of CNN-7 with full BN on CIFAR-10, with and without IBP initialization respectively, which reflects the tightness of certified bounds along the training

## B Additional Experiments

### B.1 Computational Cost for All Datasets and Models

In addition to the time cost comparison on CNN-7 on CIFAR shown in Section 4.3, we report computation cost results for all the datasets and models in Table 7. Under same training schedules, results show that our proposed method has a small overhead over vanilla IBP, and the cost is still lower than that of CROWN-IBP. Meanwhile, our method is able to achieve lower verified errors compared to the two baselines (Table 1 and Table 2). More importantly, we are able to use much shorter training schedules to achieve SOTA results compared to previous literature, which enables faster certified robust training.

### B.2 Additional Ablation Study

In this section, we present additional ablation study results on BN, where we split the centralization (the shifting operation using the mean) and the unitization (the scaling operation using the variance)

Table 7: Comparison of estimated time cost (seconds) on all the datasets and models. We report the per-epoch time during training phases with different $\epsilon$ ranges, and we report the total time when the 70-epoch schedule is used for MNIST, the 160-epoch schedule for CIFAR-10, and the 80-epoch schedule for TinyImageNet respectively. "-" in the table means that there is no $\epsilon = 0$ warmup stage for MNIST following Zhang et al. (2020). Note that on each dataset, for phases of same or different methods that are supposed to be equivalent in algorithm implementation, we make them share the same time estimation result respectively.

| Dataset | Model | Method | Per-epoch for $\epsilon$ | | | Total |
| | | | 0 | $(0, \epsilon_{\text{target}})$ | $\epsilon_{\text{target}}$ | |
|---|---|---|---|---|---|---|
| MNIST | CNN-7 | Vanilla IBP | - | 27.9 | 27.9 | 1955.1 |
| | | CROWN-IBP | - | 49.6 | 27.9 | 2387.5 |
| | | Ours | - | 37.0 | 27.9 | 2135.8 |
| | Wide-ResNet | Vanilla IBP | - | 81.0 | 81.0 | 5668.3 |
| | | CROWN-IBP | - | 142.1 | 81.0 | 6890.2 |
| | | Ours | - | 99.0 | 81.0 | 6029.3 |
| | ResNeXt | Vanilla IBP | - | 73.2 | 73.2 | 5127.2 |
| | | CROWN-IBP | - | 147.7 | 73.2 | 6616.9 |
| | | Ours | - | 104.4 | 73.2 | 5750.7 |
| CIFAR-10 | CNN-7 | Vanilla IBP | 30.0 | 54.8 | 54.8 | 8747.9 |
| | | CROWN-IBP | 30.0 | 78.5 | 54.8 | 10641.3 |
| | | Ours | 64.0 | 64.0 | 54.8 | 9512.3 |
| | Wide-ResNet | Vanilla IBP | 43.7 | 114.7 | 114.7 | 18358.4 |
| | | CROWN-IBP | 43.7 | 170.7 | 114.7 | 22764.9 |
| | | Ours | 134.7 | 134.7 | 114.7 | 19976.0 |
| | ResNeXt | Vanilla IBP | 38.7 | 102.7 | 102.7 | 16432.0 |
| | | CROWN-IBP | 38.7 | 183.3 | 102.7 | 22813.6 |
| | | Ours | 129.6 | 129.6 | 102.7 | 18611.7 |
| TinyImageNet | CNN-7 | Vanilla IBP | 282.2 | 431.4 | 431.4 | 34362.0 |
| | | CROWN-IBP | 282.2 | 663.8 | 431.4 | 36686.5 |
| | | Ours | 500.4 | 500.4 | 431.4 | 35270.3 |
| | Wide-ResNet | Vanilla IBP | 270.2 | 399.8 | 399.8 | 31861.6 |
| | | CROWN-IBP | 270.2 | 592.1 | 399.8 | 33789.3 |
| | | Ours | 464.6 | 464.6 | 399.8 | 32703.0 |
| | ResNeXt | Vanilla IBP | 197.2 | 430.5 | 430.5 | 34206.7 |
| | | CROWN-IBP | 197.2 | 883.1 | 430.5 | 38735.1 |
| | | Ours | 626.3 | 626.3 | 430.5 | 36595.8 |

Table 8: Additional ablation study results on BN where we consider whether centralization and unitization in BN present respectively. The results are from CNN-7 on CIFAR-10 ($\epsilon_{\text{target}} = 8/255$) using the training schedule with 160 epochs in total. We compare the proportion of active ReLU neurons and inactive ReLU neurons respectively, and also the errors.

| Centralization | Unitization | Active ReLU (%) | Inactive ReLU (%) | Standard error (%) | Verified error (%) |
|---|---|---|---|---|---|
| × | × | 7.37±0.25 | 90.57±0.30 | 57.36±0.45 | 69.91±0.31 |
| ✓ | × | 13.48±0.22 | 84.73±0.26 | 55.36±0.17 | 68.07±0.02 |
| × | ✓ | 16.94±0.79 | 80.40±0.75 | 54.41±0.49 | 67.78±0.46 |
| ✓ | ✓ | 21.30±0.39 | 75.90±0.40 | 51.72±0.40 | 65.58±0.30 |

to investigate whether both of them contribute to the improvement by BN. We run this experiment for CNN-7 on CIFAR-10 ($\epsilon_{\text{target}} = 8/255$) using the training schedule with 160 epochs in total, and we show the results in Table 8.

From our ablation results, we can observe that both centralization and unitization in BN contribute to the improvement. We conclude the benefit as follows. First, BN has inherent benefits for standard DNN training (Ioffe & Szegedy, 2015; Van Laarhoven, 2017; Santurkar et al., 2018). In addition, BN benefits IBP also because it has an effect on balancing ReLU neuron states, as our results show that when a model is trained with BN, the number of active ReLU neurons is noticeably better than the cases without BN. We found that actually both mean centralization and unitization help to balance active and inactive ReLU neurons. It is easy to understand that centralization helps balancing as it can center the bounds around zero. For unitization, we conjecture that it helps the optimization for DNN (from the acceleration or smoothing the loss landscape perspective for standard DNN training),

and this may allow the model to have a less tendency to reduce the robust loss by trivially making most neurons inactive.

## B.3 Other Perturbation Radii

In Table 9, we present results using perturbation radii other than those used in our main experiments. Here we consider $\epsilon_{\text{target}} \in \{0.1, 0.3\}$ for MNIST, and $\epsilon_{\text{target}} \in \{\frac{2}{255}, \frac{16}{255}\}$ for CIFAR-10. In particular, on MNIST models are trained with target perturbation radii $\epsilon_{\text{train}}$ larger than used for testing $\epsilon_{\text{target}}$ to mitigate overfitting – we use $\epsilon_{\text{train}} = 0.2$ when $\epsilon_{\text{target}} = 0.1$ and $\epsilon_{\text{train}} = 0.4$ when $\epsilon_{\text{target}} = 0.3$ following Zhang et al. (2020). We use CNN-7 in this experiment. Results show that improvements over Vanilla IBP and CROWN-IBP are consistent as in Table 1. Note that CIFAR-10 with very small $\epsilon = \frac{2}{255}$ is a special case where using pure linear relaxation bounds (Wong & Kolter, 2018; Zhang et al., 2020) for training yields even lower errors than IBP (Gowal et al., 2018) and standard CROWN-IBP which anneals to IBP training after warmup. On this setting, an alternative version of CROWN-IBP that does not anneal to IBP training can achieve lower verified error 43.61% without loss fusion (60.44% if loss fusion is enabled). However, using pure linear relaxation bounds for certified training is more costly and usually has worse results on other settings (Jovanović et al., 2021). Thus for all the other settings in Zhang et al. (2020), CROWN-IBP still have to anneal to IBP training, as the version we adopt in our main experiments. Overall, the experimental results demonstrate that our proposed method is effective on settings with different perturbation radii, compared to vanilla IBP and CROWN-IBP.

Table 9: The standard errors (%) and verified errors (%) of a CNN-7 model trained with different methods on other perturbation radii not included in the main results.

| Dataset | Warmup | $\epsilon_{\text{target}}$ | $\epsilon_{\text{train}}$ | Vanilla IBP | | CROWN-IBP | | Ours | |
|---|---|---|---|---|---|---|---|---|---|
| | | | | Standard | Verified | Standard | Verified | Standard | Verified |
| MNIST | 0+20 | 0.1 | 0.2 | 1.12 | 2.17 | 1.07 | 2.17 | 1.16 | **2.05** |
| | | 0.3 | 0.4 | 2.74 | 7.61 | 2.88 | 7.55 | 2.33 | **6.90** |
| CIFAR-10 | 1+80 | 2/255 | | 33.65 | 48.75 | 34.09 | 48.28 | 33.16 | **47.15** |
| | | 16/255 | | 64.52 | 76.36 | 71.75 | 79.43 | 63.35 | **75.52** |

## B.4 Sensitivity on the $\lambda_0$ Hyperparameter

To test the sensitivity of the training performance on the choice of $\lambda_0$, we run an experiment on CNN-7 for CIFAR-10 ($\epsilon_{\text{target}} = 8/255$) using the 160-epoch schedule. We consider $\lambda_0 \in \{0.1, 0.2, 0.5, 1.0, 2.0\}$, and we run 5 repeated experiments for each setting to report the mean and standard deviation. We show the results in Table 10.

We find that $\lambda_0 = 0.5$ or $\lambda_0 = 1.0$ both yield good results on this setting. Actually, for all the results of "ours" in Table 1 (MNIST and CIFAR-10) in the paper, we always use $\lambda_0 = 0.5$ for all settings, and we do not tune $\lambda_0$ for each setting individually. This suggests that potential users do not need to search for $\lambda_0$ in each training. Similarly, on TinyImageNet, good results can be achieved by using $\lambda_0 = 0.1$ for all the settings. The $\lambda_0$ for TinyImageNet is smaller, and this can be explained by smaller $\epsilon_{\text{target}}$ for TinyImageNet (1/255) compared to 0.4 for MNIST and $8/255$ for CIFAR-10. Thus, the results suggest that our approach is not very sensitive to choice of $\lambda_0$, and a reasonable default value can work well for many settings (e.g., under many different training schedules or models).

Table 10: Results of the sensitivity test for the $\lambda_0$ hyperparamter, on CNN-7 for CIFAR-10 ($\epsilon_{\text{target}} = 8/255$) using the 160-epoch schedule.

| $\lambda_0$ | 0.1 | 0.2 | 0.5 | 1.0 | 2.0 |
|---|---|---|---|---|---|
| Standard error (%) | $53.03 \pm 0.56$ | $53.08 \pm 0.62$ | $51.72 \pm 0.40$ | **$50.98 \pm 0.33$** | $53.80 \pm 0.37$ |
| Verified error (%) | $66.44 \pm 0.24$ | $66.54 \pm 0.48$ | **$65.58 \pm 0.32$** | $65.42 \pm 0.22$ | $66.91 \pm 0.26$ |

## B.5 Applying the Proposed Method to CROWN-IBP

We have tried applying our method to CROWN-IBP (Zhang et al., 2020) besides IBP. On CNN-7 for CIFAR-10 ($\epsilon_{\text{target}} = 8/255$) with the 160-epoch training schedule, we observe that adding BN improves the performance of CROWN-IBP (verified error $68.02\% \rightarrow 66.93\%$ if loss fusion is disabled; $76.11\% \rightarrow 68.8\%$ if loss fusion is enabled). However, further adding IBP initialization or the warmup regularizers does not significantly change the performance. For the possible reasons of this result, we analyze that: 1) CROWN-IBP already has tighter bounds by a linear relaxation based bound propagation; 2) CROWN-IBP has tight relaxation for both inactive and active ReLU neurons, compared to IBP which has tight relaxation only for inactive neurons but not active ones, so the imbalanced ReLU issue is less significant for CROWN-IBP (For the setting in Figure 2, we empirically find that even if we do not use warmup regularization, CROWN-IBP already has around 19% active neurons, even more than our improved IBP). Thus, it is reasonable that our proposed method focusing on improving bound tightness and ReLU neuron balance may be less effective for CROWN-IBP.

Instead, there may be other factors that limit the performance of linear relaxation based certified robust training. So far tighter linear or convex relaxation bounds (e.g., Zhang et al. (2018) or Wong & Kolter (2018)) usually cannot outperform pure IBP using looser interval bounds. While Zhang et al. (2020) used linear relaxation bounds for certified training and outperformed pure IBP, their method still needs to gradually anneal to pure IBP bounds in the end of training. There are some recent works that studied the reasons behind this phenomenon. Jovanović et al. (2021) identified two properties of convex relaxations, continuity and sensitivity, that may impact training dynamics.s Lee et al. (2021) identified a factor about the smoothness of loss landscape, and they proposed to use tighter relaxation via optimizing the bounds, which may lead to more favorable loss landscapes. In terms of improving the verified errors after training, Jovanović et al. (2021) only have preliminary results on a small network for MNIST since their new relaxations require solving convex/linear programs; and we outperform Lee et al. (2021) (15.42% for MNIST $\epsilon_{\text{target}} = 0.4$; 69.70% for CIFAR-10 $\epsilon_{\text{target}} = 8/255$) with a notable margin, while we use much shorter training schedules.

## B.6 Comparison with Randomized Smoothing

In this section, we empirically compare the performance of our method with randomized smoothing methods. As we have mentioned in Sec. 2, randomized smoothing is mostly suitable for $\ell_2$-norm certified defense, and it is fundamentally limited for $\ell_\infty$ norm robustness. But there are still existing works such as Salman et al. (2019) that use norm inequalities to convert $\ell_2$ norm robustness certificates to an $\ell_\infty$ norm one. For $\ell_\infty$-norm perturbation radius $\epsilon = \frac{8}{255}$ on CIFAR-10 where each input image has $3 \times 32 \times 32$ dimensions, we can convert it to $\ell_2$-norm radius $\epsilon_2 = \frac{8}{255} \times \sqrt{3 \times 32 \times 32} = 1.73884$ used by randomized smoothing such that the certified accuracy under this $\ell_2$ perturbation provides a lower bound for $\ell_\infty$ certified robustness under radius $\epsilon = \frac{8}{255}$. In an earlier work (Li et al., 2019), their certified accuracy under this perturbation size is 0, according to their Figure 1; and in a more recent work (Salman et al., 2019), according to results in their Table 7, their best certified accuracy is 23% for radius 1.75, and the certified accuracy is 26% for radius 1.5, so their certified error is at least 74% for $\ell_\infty \epsilon = 8/255$. Therefore, the certified error we achieve in our paper is much lower (65.58±0.32%), compared to randomized smoothing by converting $\ell_2$ certified radius.

## B.7 ReLU Imbalance with Shorter Warmup Length

In Figure 2, we show two 7-layer CNN models with different warmup length respectively, and the model tends to have more inactive neurons and thus more severe imbalance in ReLU neuron states for shorter warmup length, as previously mentioned in Section 3.2.2.

## B.8 Using IBP Initialization when Bound Explosion is More Severe

In Figure 7, we show that for a ResNeXt on TinyImageNet, where the explosion of certified bounds is more severe if the network is initialized with standard weight initialization, using our proposed initialization is helpful for reaching lower verified errors especially at early epochs.

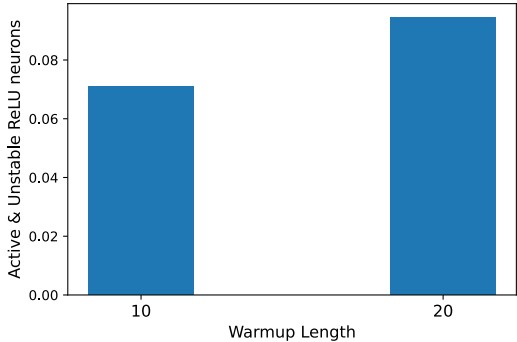

Figure 6: Ratio of active and unstable neurons in CNN-7 trained with Vanilla IBP using different warmup lengths respectively.

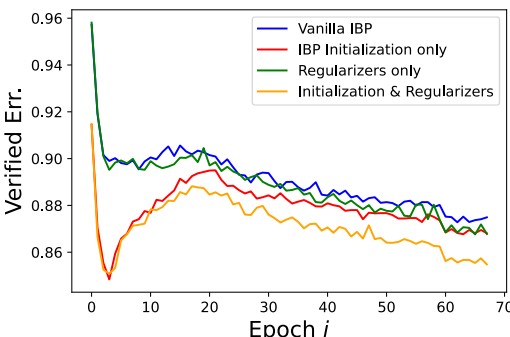

Figure 7: Curve of training verified error of a ResNeXt model on TinyImageNet. Note that the verified errors can increase during the warmup as $\epsilon$ increases.

## C   Experimental Details

**Implementation**   Our implementation is based on the `auto_LiRPA` library (Xu et al., 2020)[1] which supports robustness verification and training on general computational graphs. Baselines including Vanilla IBP and CROWN-IBP with loss fusion are inherently supported by the library. We add to implement our IBP initialization and warmup with regularizers for fast certified robust training.

**Datasets**   For MNIST and CIFAR-10, we load the datasets using `torchvision.datasets`[2] and use the original data splits. On CIFAR-10, we use random horizontal flips and random cropping for data augmentation, and also normalize input images, following Zhang et al. (2020); Xu et al. (2020). For TinyImageNet, we download the dataset from Stanford CS231n course website[3]. Similar to CIFAR-10, we also use data augmentation and normalize input images for TinyImageNet. Unlike Xu et al. (2020) which cropped the $64 \times 64$ original images into $56 \times 56$ and used a central $56 \times 56$ cropping for test images, we pad the cropped training images back to $64 \times 64$ so that we do not need to crop test images. We use the validation set for testing since test images are unlabelled, following Xu et al. (2020).

**Models**   We use three model architectures in the experiments: a 7-layer feedforward convolutional network (CNN-7), Wide-ResNet (Zagoruyko & Komodakis, 2016) and ResNeXt (Xie et al., 2017). All the models have a hidden fully-connected layer with 512 neurons prior to the classification layer. For CNN-7, there are five convolutional layers with $64, 64, 128, 128, 128$ filters respectively. For Wide-ResNet, there are 3 wide basic blocks, with a widen factor of 8 for MNIST and CIFAR-10 and 10 for TinyImageNet. For ResNeXt, we use $1, 1, 1$ blocks for MNIST and CIFAR-10, and $2, 2, 2$

---

[1]`https://github.com/KaidiXu/auto_LiRPA`

[2]`https://pytorch.org/vision/0.8/datasets.html`

[3]`http://cs231n.stanford.edu/TinyImageNet-200.zip`

blocks for TinyImageNet; the cardinality is set to 2, and the bottleneck width is set to 32 for MNIST and CIFAR-10 and 8 for TinyImageNet. For all the models, ReLU is used as the activation. These models were similarly adopted in Xu et al. (2020). But we fully add BNs after each convolutional layer and fully-connected layer, while some of these BNs were missed in Xu et al. (2020). For example, the CNN-7 model in Xu et al. (2020) had BN for convolutional layers but not the fully-connected layer. Besides, we remove the average pooling layer in Wide-ResNet as we find it harms the performance of all the considered training methods, and this modification makes the Wide-ResNet align better with the CNN-7 model, which does not have average pooling either and achieves best results compared to other models (Table 1 and Table 2).

**Training** During certified training, models are trained with Adam (Kingma & Ba, 2014) optimizer with an initial learning rate of $5 \times 10^{-4}$, and there are two milestones where the learning rate decays by 0.2. We determine the milestones for learning rate decay according to the training schedule and the total number of epochs, as shown in Table 11. Gradient clipping threshold is set to 10.0. We train the models using a batch size of 256 on MNIST, and 128 on CIFAR-10 and TinyImageNet. The tolerance value $\tau$ in our warmup regularization is fixed to 0.5. For Vanilla IBP and IBP with our initialization and regularizers, we train the models on a single NVIDIA GeForce GTX 1080 Ti or NVIDIA GeForce RTX 2080 Ti GPU for each setting. For CROWN-IBP, we train the models on two GPUs for efficiency, while in time estimation we still use one single GPU for fair comparison. The number of training and evaluation runs is 1 for each experiment result respectively. In the evaluation, the major metric is *verified error*, which stands for the rate of test examples such that the model cannot certifiably make correct predictions given the $\ell_\infty$ perturbation radius. For reference, we also report *standard error*, which is the standard error rate where no perturbation is considered.

Table 11: Milestones for learning rate decay when different total number of epochs are used. "Decay-1" and "Decay-2" denote the two milestones respectively when the learning rate decays by a factor of 0.2.

| Dataset | Total epochs | Decay-1 | Decay-2 |
|---------|--------------|---------|---------|
| MNIST | 50 | 40 | 45 |
| | 70 | 50 | 60 |
| CIFAR-10 | 70 | 50 | 60 |
| | 160 | 120 | 140 |
| TinyImageNet | 80 | 60 | 70 |

**Warmup scheduling** During the warmup stage, after training with $\epsilon = 0$ for a number of epochs, the perturbation radius $\epsilon$ is gradually increased from 0 until the target perturbation radius $\epsilon_{\text{target}}$, during the $0 < \epsilon < \epsilon_{\text{target}}$ phase. Specifically, during the first 25% epochs of the $\epsilon$ increasing stage, $\epsilon$ is increased exponentially, and after that $\epsilon$ is increased linearly. In this way, $\epsilon$ remains relatively small and increases relatively slowly during the beginning, to stabilize training. We use the `SmooothedScheduler` in the `auto_LiRPA` as the scheduler for $\epsilon$ similarly adopted by Xu et al. (2020). On CIFAR-10, unlike some prior works which made the perturbation radii used for training 1.1 times of those for testing respectively (Gowal et al., 2018; Zhang et al., 2020), we find this setting makes little improvement over using same perturbation radii for both training and testing in our experiments as also mentioned in Lee et al. (2021), and thus we directly adopt the later setting for simplicity.

# D  Mathematical Proofs

## D.1  Proof of Eq. (5)

In this section, we provide a proof for Eq. (5):

$$\mathbb{E}(\delta_i) = \mathbb{E}(\text{ReLU}(\overline{\mathbf{h}}_i)) - \text{ReLU}(\underline{\mathbf{h}}_i)) = \frac{1}{2}\mathbb{E}(\Delta_i), \tag{11}$$

where $\Delta_i = \overline{\mathbf{h}}_i - \underline{\mathbf{h}}_i$, and $\delta_i = \overline{\mathbf{z}}_i - \underline{\mathbf{z}}_i$.

*Proof.* We first have

$$\mathbb{E}(\delta_i) = \mathbb{E}(\text{ReLU}(\overline{\mathbf{h}}_i) - \text{ReLU}(\underline{\mathbf{h}}_i))$$

$$= \mathbb{E}(\text{ReLU}(\mathbf{c}_i + \frac{\Delta_i}{2}) - \text{ReLU}(\mathbf{c}_i - \frac{\Delta_i}{2})) \tag{12}$$

$$= \mathbb{E}(\text{ReLU}(\mathbf{c}_i + \frac{\Delta_i}{2})) - \mathbb{E}(\text{ReLU}(\mathbf{c}_i - \frac{\Delta_i}{2})).$$

Note that $\mathbf{c}_i = \frac{1}{2}\mathbf{W}_i(\underline{\mathbf{z}}_i + \overline{\mathbf{z}}_i)$ and $\Delta_i = |\mathbf{W}_i|\delta_i$, and thus $p(-\mathbf{c}_i \mid |\mathbf{W}_i|) = p(\mathbf{c}_i \mid |\mathbf{W}_i|)$ and $p(-\mathbf{c}_i|\Delta_i) = p(\mathbf{c}_i|\Delta_i)$, where we use $p(\cdot)$ to denote the probability density function (PDF). Thereby,

$$\mathbb{E}(\text{ReLU}(\mathbf{c}_i + \frac{\Delta_i}{2})) = \int_0^\infty \int_{-\frac{\Delta_i}{2}}^\infty (\mathbf{c}_i + \frac{\Delta_i}{2})p(\mathbf{c}_i|\Delta_i)p(\Delta_i)d\mathbf{c}_i d\Delta_i,$$

$$\mathbb{E}(\text{ReLU}(\mathbf{c}_i - \frac{\Delta_i}{2})) = \int_0^\infty \int_{\frac{\Delta_i}{2}}^\infty (\mathbf{c}_i - \frac{\Delta_i}{2})p(\mathbf{c}_i|\Delta_i)p(\Delta_i)d\mathbf{c}_i d\Delta_i. \tag{13}$$

And thus

$$\mathbb{E}(\text{ReLU}(\mathbf{c}_i + \frac{\Delta_i}{2})) - \mathbb{E}(\text{ReLU}(\mathbf{c}_i - \frac{\Delta_i}{2}))$$

$$= \int_0^\infty (\int_{\frac{\Delta_i}{2}}^\infty \Delta_i + \int_{-\frac{\Delta_i}{2}}^{\frac{\Delta_i}{2}} (\mathbf{c}_i + \frac{\Delta_i}{2}))p(\mathbf{c}_i|\Delta_i)p(\Delta_i)d\mathbf{c}_i d\Delta_i$$

$$= \int_0^\infty \int_{-\infty}^\infty \frac{\Delta_i}{2}p(\mathbf{c}_i|\Delta_i)p(\Delta_i)d\mathbf{c}_i d\Delta_i \tag{14}$$

$$= \frac{1}{2}\mathbb{E}(\Delta_i).$$

$\square$

## D.2   Proof on the Bounds of $\text{Var}(\underline{\mathbf{h}}_i)$ and $\text{Var}(\overline{\mathbf{h}}_i)$

In this section, we show that $\text{Var}(\underline{\mathbf{h}}_i)$ and $\text{Var}(\overline{\mathbf{h}}_i)$ will not explode or vanish at initialization, so that the magnitude of forward signals will not vanish or explode when we use IBP initialization which focuses on stabilizing the tightness of certified bounds.

We can derive that

$$\text{Var}(\overline{\mathbf{h}}_i) = \text{Var}(\mathbf{W}_{i,+}\overline{\mathbf{z}}_{i-1} + \mathbf{W}_{i,-}\underline{\mathbf{z}}_{i-1})$$

$$= \text{Var}([\mathbf{W}_{i,+}\overline{\mathbf{z}}_{i-1} + \mathbf{W}_{i,-}\underline{\mathbf{z}}_{i-1}]_j) \ (0 \le j \le r_i)$$

$$= \text{Var}\Big(\sum_{k=1}^{n_i}([\mathbf{W}_i]_{j,k}[\overline{\mathbf{z}}_{i-1}]_k \cdot \mathbb{I}([\mathbf{W}_i]_{j,k} > 0))$$

$$+ \sum_{k=1}^{n_i}([\mathbf{W}_i]_{j,k}[\underline{\mathbf{z}}_{i-1}]_k \cdot \mathbb{I}([\mathbf{W}_i]_{j,k} \le 0))\Big).$$

Since $\mathbf{W}_i$ is initialized with mean 0, the numbers of negative elements and positive elements are approximately equal, and thus

$$\text{Var}(\overline{\mathbf{h}}_i) \approx \frac{n_i}{2}\text{Var}(\mathbf{W}_{i,+}\overline{\mathbf{z}}_{i-1}) + \frac{n_i}{2}\text{Var}(\mathbf{W}_{i,-}\underline{\mathbf{z}}_{i-1})$$

$$= \frac{n_i}{2}\Big(\text{Var}(\mathbf{W}_{i,+})\mathbb{E}(\overline{\mathbf{z}}_{i-1})^2$$

$$+ \text{Var}(\overline{\mathbf{z}}_{i-1})\mathbb{E}(\mathbf{W}_{i,+})^2 + \text{Var}(\mathbf{W}_{i,-})\mathbb{E}(\underline{\mathbf{z}}_{i-1})^2 + \text{Var}(\underline{\mathbf{z}}_{i-1})\mathbb{E}(\mathbf{W}_{i,-})^2\Big)$$

$$= \frac{\pi}{n_i}(1 - \frac{2}{\pi})\mathbb{E}(\overline{\mathbf{z}}_{i-1}^2) + \frac{2}{n_i}\text{Var}(\overline{\mathbf{z}}_{i-1}) + \frac{\pi}{n_i}(1 - \frac{2}{\pi})\mathbb{E}(\underline{\mathbf{z}}_{i-1}^2) + \frac{2}{n_i}\text{Var}(\underline{\mathbf{z}}_{i-1}).$$

Note that $\mathbb{E}(\overline{\mathbf{z}}_i) \ge \mathbb{E}(\delta_i)$ and we have made $\mathbb{E}(\delta_i)$ stable in each layer. Thus $\text{Var}(\overline{\mathbf{h}}_i) \ge \frac{n_i}{2}\text{Var}(\mathbf{W}_{i,+})\mathbb{E}(\overline{\mathbf{z}}_{i-1})^2$ and will not vanish when the network goes deeper. Also note that $n_i > 1$ in neural networks, and therefore $\text{Var}(\overline{\mathbf{h}}_i)$ will not explode. The same analysis can also be applied to $\underline{\mathbf{h}}_i$.

However, when we use the IBP initialization, variance of the standard forward value $\mathbf{h}_i$ will be smaller than that of Xavier and Kaiming Initialization. Following the analysis in He et al. (2015a), we have

$$\text{Var}(\mathbf{h}_i) = \frac{n_i}{2}\text{Var}(\mathbf{W}_i)\text{Var}(\mathbf{h}_{i-1}).$$

In IBP initialization, we have $\text{Var}(\mathbf{W}_i) = \frac{2\pi}{n_i^2}$, and the variance of $\mathbf{h}_i$ can become smaller after going through each affine layer. Therefore, as mentioned in Section 4.4, simply adding IBP initialization may not finally improve the verified error, because it may harm the early warmup when $\epsilon$ is small and certified training is close to standard training. In this paper, in addition to IBP initialization, we further add regularizers to stabilize certified bounds and the balance of ReLU neuron states, while the variance is stabilized by fully adding BN. The effect of these parts of our proposed method is discussed in Section 4.4.