# OpenReview forum: "Fast Certified Robust Training with Short Warmup"
_NeurIPS.cc/2021/Conference — NeurIPS 2021 Poster_

### Official Review · Reviewer_Myoo · 2021-07-13

**Rating:** 6
**Confidence:** 3

**Summary:**

This work analyzes the possible reasons why extremely large warmup and training epochs are necessary in former IBP training methods. Then it proposes a new training method for fast certified robust training. Comparing with its counterparts, the proposed method can obtain SOTA performance within much fewer training epochs. The experiments prove the efficiency of the proposed method.


**Ethical Concerns:**

No.

**Limitations And Societal Impact:**

Yes.

**Main Review:**

Originality: The proposed method is kind of incremental (BN + two additional loss); the analysis on the concerned issues does provide some interesting insights, but the motivations/intuition of proposed method are not reasonable.

significance: The performance of proposed method outperforms its counterparts with a clear margin, but the improvement is kind of minor (performance shown in table 1, 2, with same short schedule).

Quality: The experiment results shown in the paper are clear and full of details, it is very convenient for me to assess the capacity of the proposed method. But the analysis part is unsatisfactory.

Detailed comments: I think the most important contribution of this work is to explain why former IBP training methods require extremely large training epochs, and design appropriate strategies to solve it. As far as I am concerned, the interpretations on issues (Sec 3.2) are reasonable. But the motivation to use BN (section 3.3.2) is not convincing, because it will destroy the foundation of interval bound propagation framework: note according to section 3.2.1, the interval bound can be estimated because the network only consists of linear transformation and monotone activation function. The derivation in section 3.2.1 cannot be held in normalized network. I think this might be the core reason why previous works do not utilize BN too much. Since BN is an important part of the proposed method, the author should take the effect of BN into account when demonstrate their intuition of the method. Besides, in section 3.3.2, the authors states BN is beneficial to IBP mostly by improving the balance of ReLU activation, they do not discuss another characteristic of BN, accelerating training of DNN [1,2]. It is very likely that the proposed method can be trained within fewer epochs due to accelerating ability of BN rather than balancing ReLU activation. The authors should carefully review the property of BN and clarify what is the intrinsic benefits brought by BN to IBP.

Suggestions: 1) To verify the benefits of BN, extra ablation study should be enough because BN can be dissected into two parts: centralization (minus mean, balancing ReLU activation) and unitization (divided by standard deviation). Centralization contributes to balancing effect while unitization contribute to acceleration effect. 2) But it's really important to clarify how BN affects the interval bound in the propagation network in Section 3.2.1.

Clarity: the writing is good, I enjoy reading the paper.

[1] Santurkar S, Tsipras D, Ilyas A, et al. How does batch normalization help optimization?[C]//Proceedings of the 32nd international conference on neural information processing systems. 2018: 2488-2498.

[2] Van Laarhoven T. L2 regularization versus batch and weight normalization[J]. arXiv preprint arXiv:1706.05350, 2017.


=====

Edit after seeing author's rebuttal

After seeing the response the author's rebuttal, I'm still not satisfied with the author's interpretations on how BN affects the interval bound in the propagation network, because they only add extra empirical verifications rather than provide solid theoretical demonstrations, I think the authors need more efforts to clarify this issues in their next version no matter whether this paper is accepted. I appreciate the addition ablation study on BN, so I raise my score to 6.



**Time Spent Reviewing:**

6 hours

---

> ### Author Response · Authors · 2021-08-11
> **We provide additional ablation study and analysis for BN layers**
>
> We thank the reviewer for the very constructive comments, especially the ones related to batch normalization. Following your suggestions, we provide additional analysis for using BN with IBP as well as empirical ablation studies as you requested. Your suggestions helped us rethink BN and are very valuable. Here we answer your questions below:
> ### Using BN on IBP and its analysis
> First of all, we want to clarify that adding BN in IBP training actually does not destroy IBP computation. BN has been partly used in some prior works that also involve IBP (Wong et al., 2018; Xu et al., 2020), and when BN presents, IBP computation still holds. But we agree with the reviewer that we need to include an analysis of IBP with BN, which is presented below.
>
> Now we analyze the tightness of certified bounds when BN presents. As we mentioned in Line 181~183, we use mean and variance estimation computed from clean data in BN (which is also the standard way). After propagating IBP bounds $ \underline{\boldsymbol{h}}_i $ and $ \overline{\boldsymbol{h}}_i$ through BN, we use $\underline{\boldsymbol{h}}_i'$ and $\overline{\boldsymbol{h}}_i'$ to denote the output after BN.
> To propagate the bounds through BN, we have  $\underline{\boldsymbol{h}}_i' = \alpha_i \frac{\underline{\boldsymbol{h}}_ii - \mu(\boldsymbol{h}_i)}{\sigma(\boldsymbol{h}_i)} + \beta_i $ (when $\alpha_i$ > 0),
> where $\mu(\boldsymbol{h}_i)$ and $\sigma(\boldsymbol{h}_i) $ stand for the estimated mean and variance respectively from clean output $ \boldsymbol{h}_i $. Similarly we can get $\overline{\boldsymbol{h}}_i'$. Therefore, to conduct a analysis similar to Sec. 3.2.1 for BN, we first need to estimate $\mu(\boldsymbol{h}_i)$ and $\sigma(\boldsymbol{h}_i)$, and then we can estimate $\overline{\boldsymbol{h}}_i', \underline{\boldsymbol{h}}_i'$. Finally, we have $\Delta_i' = \overline{\boldsymbol{h}}_i' - \underline{\boldsymbol{h}}_i'$ to denote the bound tightness after BN.
>
> After the neural network is initialized, we can assume elements in $\boldsymbol{h}_i$ are initialized to independent Gaussian variables and the mean and variance can be quantified to calculate $\Delta_i’$. However, after a single step of optimization, elements in $\boldsymbol{h}_i$ are not independent and the mean and variance are difficult to calculate explicitly, but they can be estimated empirically. Although when $\sigma(\boldsymbol{h}_i) < 1$, $\Delta_i’$ will get larger than $\Delta_i$ (bounds become looser after propagating through BN), we can show empirically that IBP initialization is still able to tighten the bounds with BN, as demonstrated below.
>
> In the table below, we compare $\log(\mathbb{E}(\Delta_L)/\mathbb{E}(\Delta_0))$ of a CNN-7 model with BN on CIFAR, with and without IBP initialization respectively, during the early epochs. A smaller value indicates that the bounds are tighter. And we can see that the model with IBP initialization has smaller $\mathbb{E}(\Delta_L)/\mathbb{E}(\Delta_0)$ and thus tighter bounds. We will add the above analysis about BN in our later revision.
>
> | Initialization | Epoch 1 | Epoch 2 | Epoch 3 | Epoch 4 |
> | -------------- | ------- | ------- | ------- | ------- |
> | No             | 16.29   | 15.21   | 13.08   | 11.90   |
> | Yes            | 11.56   | 12.42   | 11.97   | 11.24   |
> ### Ablation study about BN
> Thank you for this nice suggestion. We have added additional ablation study as suggested. The results for CIFAR-10 with 160 total training epochs are shown in the table below, where “BN-Mean” and “BN-Var” stand for whether we enable the mean term (centralization) and variance term (unitization) respectively in BN.
>
>
> | BN-Mean | BN-Var | Active ReLU (%) | Inactive ReLU (%) | Standard error (%) | Verified error (%) |
> | ------- | ------ | --------------- | ----------------- | ------------------ | ------------------ |
> | No      | No     | 7.37±0.25       | 90.57±0.30        | 57.36±0.45         | 69.91±0.31         |
> | Yes     | No     | 13.48±0.22      | 84.73±0.26        | 55.36±0.17         | 68.07±0.02         |
> | No      | Yes    | 16.94±0.79      | 80.40±0.75        | 54.41±0.49         | 67.78±0.46         |
> | Yes     | Yes    | 21.30±0.39      | 75.90±0.40        | 51.72±0.40         | 65.58±0.30         |
>
>
> From our ablation results, we can observe that both centralization and unitization contribute to the performance improvement. We conclude the benefit as follows. First, BN has inherent benefits for standard DNN training  (e.g., acceleration mentioned by the reviewer, and also smoothing the loss landscape as mentioned by Reviewer E79q). In addition, BN benefits IBP also because it has an effect on balancing ReLU neuron states in certified defense, as our results show that when a model is trained with BN, the number of active ReLU neurons is noticeably better than the cases without BN. We found that actually both centralization and unitization help to balance active and inactive ReLU neurons. It is easy to understand that centralization helps balancing as it can center the bounds around zero. For unitization, it helps the optimization for DNN (from the acceleration or loss landscape perspective), and this may allow the model to have a less tendency to reduce the robust loss by trivially making most neurons inactive.
>
> In our paper, we will revise our discussion for the contribution of BN to IBP training according to your suggestions, including a subsection in Relate Work to cite and discuss relevant papers, and also include the ablation study results. The list of papers provided by the reviewer is really helpful for us.
>
> Finally, we would like to thank the reviewer again for the very constructive comments. We hope the new analysis and results provided by us are helpful, and please kindly let us know if you have any additional questions or concerns.

---

> ### Author Response · Authors · 2021-08-27
> **Thank you again for the insightful review, and hope you can reevaluate our paper**
>
>
> Dear Reviewer Myoo,
>
> We are very grateful for your insightful review. We will really appreciate it if you could kindly take a look at our response because the discussion period is ending soon.
>
> In our response, we have provided a new analysis for IBP with BN, and additional ablation studies on BN as you requested. Your insight on decomposing BN into centralization and unitization is very helpful and is also well aligned with our observations. We will cite the papers you mentioned and include more discussions on BN in related work.
>
> We hope the reviewer can reevaluate our paper based on our full response, and please kindly let us know if you have any further questions or comments on our paper.
>
> Sincerely,
> Paper 8739 Authors

---

### Official Review · Reviewer_E79q · 2021-07-15

**Rating:** 7
**Confidence:** 3

**Summary:**

This paper aims at improving upon the training time of existing certified robust training methods. Existing methods suffer from long warm-up time needed to make the training dynamics of robust training methods stable. Specifically, the authors identify two drawbacks in existing methods, viz, poor parameter initialization and an imbalance in hidden layers' ReLU activations, both of which are shown to make the training dynamics of certified robust methods unstable. The authors propose several fixes to the existing Interval Bound Propagation (IBP) including a weight initialization, regularizations and recommend using batch norm in all layers to stabilize IBP training thereby making the warmup phase of training shorter. They show that their method achieves competitive standard/verified accuracy while being able to train significantly faster.





**Limitations And Societal Impact:**

See main review for limitations. I think given the paper is about adversarial robustness, a short discussion on societal impact would be appropriate.

**Main Review:**


strengths:
- paper is clearly written and the ideas were easy to follow
- the proposals made by the authors were explained in detail and immediately made sense
- experiments show substantial improvement in training speed using the proposed technique
- the ablation analysis clearly shows the importance of each of the proposed components that are added to vanilla IBP

weakness:
- the proposed method introduces regularization terms which requires the regularization coefficient lambda in Eq 10. Since the main claim is regarding the speedup in training time, I think a discussion on how sensitive the performance is w.r.t. lambda would be informative. If the performance is too sensitive and extensive tuning is required, then it defeats the purpose of training time speedup. For instance, if we even perform a grid search over 5 values, the combined training time for the proposed method on CIFAR-10 (based on Table 3) would be roughly 47000 sec, which would not be a fair comparison with the baselines. Ideally I would like to see something like a table with a computational budget and the best performance along with training time of the proposed method.
- deriving initialization schemes for non-MLP (and CNN) type architectures such as ResNets requires additional considerations and cannot be trivially extended from MLP/CNN style weight initialization schemes (cf [1], [2]). Since the authors use ResNet architecture in this paper, I think that while the proposed initialization improves training speed, it may still not be the ideal initialization. This should at least be discussed in the paper.
- In section 3.3.2, it is mentioned that BN accelerates convergence by reducing internal covariate shift. This is an outdated view. More recent research suggest BN instead improves the smoothness of the loss landscape (cf [3]).

Overall I like the paper. If the authors address my concerns, I will keep my score.

Minor comments:
- Line 43: grammatical error. "can be exploded" should be "explode"
- Line 48: "during the training start" should be "at the start of training"

[1] Hanin, B., & Rolnick, D. (2018). How to start training: The effect of initialization and architecture. arXiv preprint arXiv:1803.01719.

[2] Arpit, D., Campos, V., & Bengio, Y. (2019). How to initialize your network? robust initialization for weightnorm & resnets. arXiv preprint arXiv:1906.02341.

[3] Santurkar, S., Tsipras, D., Ilyas, A., & Mądry, A. (2018, December). How does batch normalization help optimization?. In Proceedings of the 32nd international conference on neural information processing systems (pp. 2488-2498).

**Time Spent Reviewing:**

8 hours

---

> ### Author Response · Authors · 2021-08-11
> **We add discussions on the choices of $\lambda$, initialization for ResNet architectures, and the benefit of BN**
>
> We thank the reviewer for recognizing the contributions of our paper and valuable comments. We address the concerns below, and we will add the discussions in our revision.
>
> ### About $\lambda$
> On CIFAR-10 with 160 training epochs, we tried different $\lambda$ values in $ \{0.1,0.2,0.5,1.0,2.0 \}$. As the table below shows, we find that $\lambda=0.5$ or $\lambda=1.0$ both yield good results on this setting. Actually, for all the results of “ours” in Table 1 (MNIST and CIFAR-10) in the paper, we always use $\lambda=0.5$ for all settings, and we do not tune $\lambda$ for each setting individually. This suggests that potential users do not need to search for $\lambda$ by themselves for each training.
>
> | $\lambda$          | 0.1          | 0.2          | 0.5            | 1.0              | 2.0          |
> | ------------------ | ------------ | ------------ | -------------- | ---------------- | ------------ |
> | Standard error (%) | 53.03 ± 0.56 | 53.08 ± 0.62 | **51.72±0.40** | **50.98 ± 0.33** | 53.80 ± 0.37 |
> | Verified error (%) | 66.44 ± 0.24 | 66.54 ± 0.48 | **65.58±0.32** | **65.42 ± 0.22** | 66.91 ± 0.26 |
>
> Similarly, on TinyImageNet, good results can be achieved by using $\lambda=0.1$ for all the settings. The $\lambda$ for TinyImageNet is smaller, and this can be explained by smaller $\epsilon$ for TinyImageNet (1/255) compared to 0.4 for MNIST and 8/255 for CIFAR-10. We will add discussions on the choice of $\lambda$ in our revision.
>
> Overall, we believe our approach is not very sensitive to the lambda regularizer, and a reasonable default can actually work well in many situations (e.g., under many different training schedules). We also did not extensively tune this hyperparameter in our experiments. We hope these results can address your concern.
>
> ### Initialization for ResNet
> We thank the reviewer for providing the related works on resnet initialization. These two papers provide the analysis of resnet initialization, but they focus on clean training and aim to keep the variance stable. In our setting, we want to make $E(\Delta_i)$ stable. And we can consider a similar setting as in Arpit et al., 2019 where $x_l = x_{l-1} + F(x_{l-1})$, $F(\cdot)$ is a feed-forward network and we already keep $\Delta(x_{l-1}) = \Delta(F(x_{l-1}))$ (we use $\Delta(\cdot)$ to represent the difference between upper and lower certified bounds). Then with the residual connection, we have $\Delta(x_l) = 2 \Delta(x_{l-1})$. There is additionally only a constant factor 2, which does not depend on width $n_l$, compared to feedforward networks, and we can also remove this factor by dividing the weight after each residual connection by 2. We will add this discussion in our revision.
>
> ### About BN
> We thank the reviewer for correcting that. We will adjust our description about BN. Regarding the loss landscape, the benefit of having a smooth loss landscape was also discussed in Lee et al. 2021 for certified defense, in a different scenario. They focus on improving linear relaxation based certified defense and propose to use tighter bounds via optimization, which may lead to a smoother loss landscape. In contrast, we focus on improving more efficient IBP training for fast training, and we propose different techniques. And we outperform Lee et al., 2021 (15.42% for MNIST eps=0.4; 69.70% for CIFAR-10 eps=8/255) with a notable margin (we get much lower verified errors 10.82% on MNIST and 65.03% on CIFAR-10, while using short training schedules). We will cite these works and modify our discussion accordingly.
>
> Lee et.al.: “Loss Landscape Matters: Training Training Certifiably Robust Models with Favorable Loss Landscape”, 2021
>
> Arpit, D., Campos, V., & Bengio, Y. (2019). How to initialize your network? robust initialization for weightnorm & resnets. arXiv preprint arXiv:1906.02341.

---

### Official Review · Reviewer_e75G · 2021-07-15

**Rating:** 6
**Confidence:** 3

**Summary:**



In this paper, the authors propose a  weight initialization scheme and regularizers for certified robust training.  Moreover, the authors argue the benefit of Batch Normalization (BN) in certified training.

**Limitations And Societal Impact:**

The limitations are stated.

**Main Review:**


Pros.
 1. The idea of keeping the difference gain close to one to achieve stable bounds is interesting.
2. The paper focuses on certified robust training, which is an important direction in robust deep learning.


Cons.
1. Some symbols are confusing.   In Eq.(4),  |\boldsymbol{W}_i| seems to be a vector. But in Line 171 E(|\boldsymbol{W}_i|) is a scalar.  How to achieve  E(\Delta_i) = n_i E(|\boldsymbol{W}_i|) E(\delta_i) in Line 132?
2. In table 2, the error is around 80%. In Figure 5, the classification error is larger than 85%.  I think in this setting, the performance is too weak to be practically useful. It is unconvinced to test in this extreme setting.  Thus, the empirical study on the only large dataset (TinyImagenet) cannot support the conclusion.

Minor:
Typos.  "last year" in Line 118.


==============================================================

==================Review  Update  ===============================

Thanks for the authors' detailed feedback. Most of my concerns have been clarified.   So I decided to raise my score to 6.  In addition,  it seems that $\mathbb{E} \Delta_i$ is determined by the difference of the worst-case samples on the boundary.  The empirical estimation $\\mathbb{E} \widehat{\Delta}_i$ using a mini-batch may have a large variance.  It is better to verify the variance of  $\\mathbb{E} \widehat{\Delta}_i$  w.r.t the batch-size.







**Time Spent Reviewing:**

3

---

> ### Author Response · Authors · 2021-08-11
> **We clarified the symbols. High error is common for certified defense especially for large datasets.**
>
> We thank the reviewer for the valuable suggestions. We feel **the criticism of high “classification error” is due to misunderstanding**. We argue that the high verified error is commonly seen in certified defense [a,b,c,d,e] due to the hardness of the problem, and is a common limitation for many certified adversarial defenses (see details below). Additionally, we will clarify the notations and symbols in our paper.
> We answer your questions in detail below:
>
> ### Performance on TinyImageNet
> In Figure 5, we presented the **verified error** which is a guaranteed **upper bound of error under any adversarial attacks**, *not* ordinary classification error. Over 80% verified error is actually already difficult to achieve on TinyImageNet:
>
> 1. **Existing SOTA methods also have high verified errors** on the TinyImageNet dataset. For example, Xu et al., 2020 (**NeurIPS 2020**) has **84.14% certified error** with 400 training epochs. Instead, our approach achieved **82.36% certified error** with only 80 training epochs. Many other certified defenses such as [a][b][c] cannot even be scaled to the TinyImageNet setting.
>
> 2. Verified error gives provable theoretical guarantees, where no adversarial attack can make the classification error worse than the verified error. For a large neural network, it is very challenging to give theoretical guarantee for its robustness, and it is the goal of certified defense [a,b,c,d,e]. An ordinarily trained classifier has 100% verified error (much worse than 80%) because it can be easily attacked to 0% accuracy.
>
> 3. TinyImageNet has 200 labels, so a random guess model would have 99.5% error. 80% is much lower than random guess.
>
> We do believe that currently the relatively high error is a limitation of existing certified defense, however it is a well-known challenge and we have outperformed SOTA works in this field. The field of certified defense is still young and Rome was not built within a day. We feel it is unfair to criticize our work due to high verified errors on this very challenging problem, and we hope our explanations and pointers to relevant works have cleared up your misunderstandings. We sincerely hope the reviewer can reevaluate our paper based on our response. Thank you.
>
> ### Clarification on the symbols
> Sorry for the confusion, we will revise our paper to make the notations more clear. Here we assume elements in $|\boldsymbol{W}_i|$ are independent variables following the same distribution at initialization. And we use $\mathbb{E}(|\boldsymbol{W}_i|)$ (a single number) to denote the expectation of this distribution, so $\mathbb{E}(|\boldsymbol{W}_i|) = \mathbb{E}(|\boldsymbol{W}(i, jk)|)$ for all $j, k$. The notation is similar for $E(\Delta_i)$ and $E(\delta_i)$.
>
> ### How to achieve $E(\Delta_i) = n_i E(|\boldsymbol{W}_i|) E(\delta_i)$ in Line 132?
>
> We apologize for this hard to understand equation, and we present a short derivation below:
> For $E(\Delta_i) = n_i E(|\boldsymbol{W}_i|) E(\delta_i)$, we note that in Eq (4), we have $\Delta_i = |\boldsymbol{W}_i| \delta_i $, therefore for an element in $\Delta_i$ which we denote as $\Delta (i,j)$, we have $\Delta (i,j) = \sum_k |\boldsymbol{W}(i, jk)| \delta(i, k)$. Since $|\boldsymbol{W}(i, jk)|$ and $\delta (i,k)$ are independent, $\mathbb{E}(|\boldsymbol{W}(i,jk)| \delta(i,k)) = \mathbb{E}(|\boldsymbol{W}(i,jk)|) \mathbb{E}(\delta(i,k))$. Also note that we defined $\mathbb{E}(|\boldsymbol{W}_i|) = \mathbb{E}(|\boldsymbol{W}(i, jk)|)$ and $ \mathbb{E}(\delta_i)= \mathbb{E}(\delta(i,k))$, then we have $E(\Delta_i) = n_i E(|\boldsymbol{W}_i|) E(\delta_i)$.
>
> Please kindly let us know if you have any additional concerns on the notation or any equations. Thank you.
>
> References:
>
> [a] Wong, E. & Kolter, Z.. (2018). Provable Defenses against Adversarial Examples via the Convex Outer Adversarial Polytope. Proceedings of the 35th International Conference on Machine Learning, in Proceedings of Machine Learning Research 80:5286-5295
>
> [b] Zhang H, Chen H, Xiao C, et al. Towards stable and efficient training of verifiably robust neural networks[J]. arXiv preprint arXiv:1906.06316, 2019.
>
> [c] Singh G, Gehr T, Mirman M, et al. Fast and Effective Robustness Certification[J]. NeurIPS, 2018, 1(4): 6.
>
> [d] Xu, K., Shi, Z., Zhang, H., Wang, Y., Chang, K.-W., Huang, M., Kailkhura, B., Lin, X., and Hsieh, C.-J. Automatic perturbation analysis for scalable certified robustness and beyond. Advances in Neural Information Processing Systems, 33, 2020.
>
> [e] Gowal S, Dvijotham K, Stanforth R, et al. On the effectiveness of interval bound propagation for training verifiably robust models[J]. arXiv preprint arXiv:1810.12715, 2018.

---

> ### Author Response · Authors · 2021-08-27
> **We hope the reviewer can check out our response because there were misunderstandings in the review**
>
> Dear Reviewer e75G,
>
> We thank you again for your helpful review. The discussion period is ending soon, and we sincerely hope you can reevaluate our paper based on our response because there were crucial misunderstandings.
>
> The high *verified error* (note that this is *not the ordinary classification error*) is common in literature due to the hardness of certified defense on large models for TinyImageNet. For example, [50] (Xu et al., NeurIPS 2020) has **84.14%** certified error after **400 epochs** training, and we already outperform it (**82.36%** certified error) using **only 80 training epochs**. Many other certified defenses *cannot even be scaled to TinyImageNet* under our setting.
>
> Additionally, in our response, we also clarified the symbols and equations and will revise our paper according to your suggestion. We hope the reviewer can reevaluate our paper based on our response. Thank you.
>
> Sincerely,
> Paper 8739 Authors

---

### Official Review · Reviewer_3WeK · 2021-07-16

**Rating:** 7
**Confidence:** 4

**Summary:**

The authors identify two issues with current certified robustness training approaches: Weight initialisation leading to exploding bounds and imbalance in ReLU activation states. They propose three improvements to mitigate these issues and cut down training time: new weight initialisation (IBP initialisation), batch normalisation after every layer and custom regularisation to encouraging tight bounds and balance ReLU activations. The approach is evaluated on MNIST, CIFAR-10 and TinyImageNet.

**Limitations And Societal Impact:**

Societal Impact was not discussed.

**Main Review:**

Overall the paper is well written and easy to understand. The investigated problem is well motivated and the solutions are natural. Further, improving and speeding up certified training is an important problem. While the approach and techniques used in the paper are reasonably simple they are quite elegant and further seem to be  effective, as shown in the experimental evaluation.

A general question i had while reading the paper, was wether the introduced methods could be applied or adapted for CROWN-IBP, since  similar gains might be achievable there.

While reading Sec. 3, i expected even bigger improvements compared to the reported results in Sec. 4, when IBP initialisation is used compared to standard initialisation as an exponential blow-up was mitigated. Could the authors report the numbers in Table 4, where no Batch-Normalisation is used but IBP initialisation with and without the regularisers is used?

Further Comments:
- Figure 3 & 4: The font size for the labels (axes) could be improved
- Appendix D.2: Equation after line 616: the brackets do not match.
- The authors may want to highlight the differences to other works considering problems with current certified robustness training methods [A, B].

[A] Lee et.al.: “Loss Landscape Matters: Training Training Certifiably Robust Models with Favorable Loss Landscape”, 2021

[B] Jovanovic et.al.: “Why Tighter Relaxations May Hurt Training”, https://arxiv.org/abs/2102.06700

**Time Spent Reviewing:**

4

---

> ### Author Response · Authors · 2021-08-11
> **We add experiments with CROWN-IBP, additional ablation study, and also discussions with related work.**
>
> ## About CROWN-IBP
> We have tried applying our improvements for IBP to CROWN-IBP. On CIFAR-10 with 160 training epochs, we observe that BN improves the performance (verified error 68.02%->66.93% if loss fusion is disabled; 76.11%->68.8% if loss fusion is enabled). However, further adding IBP initialization or the warmup regularizers does not significantly change the results.
> For the possible reasons that adding IBP initialization and warmup regularizers do not improve CROWN-IBP, we conjecture that: 1) CROWN-IBP already has tighter bounds by a linear relaxation based bound propagation; 2) CROWN-IBP has tight relaxation for both inactive and active ReLU neurons, compared to IBP which has tight relaxation only for inactive neurons but not active ones, so the imbalanced ReLU issue is less significant for CROWN-IBP (we empirically find that even if we do not use warmup regularization, CROWN-IBP has around 19% active neurons, even more than “ours” using improved IBP in Figure 2). Thus, it is reasonable that improvements focusing on bound tightness and ReLU neuron balance may be less effective for CROWN-IBP. There might be other factors that limit the performance of linear relaxation based certified training (as partly addressed in the papers mentioned by the reviewer) and this may be a direction for future work.
> Overall, with short training schedules, our improved IBP outperforms CROWN-IBP under the same schedule. Besides, CROWN-IBP is inherently more costly than IBP per epoch. So our improved IBP is good for fast certified robust training.
>
> ## Ablation study
> In the table below, we report additional ablation study results on CIFAR-10 with 160 total training epochs. Simply adding the initialization has a limited effect which is weakened during the warmup stage with relatively small $\epsilon$, but initialization can still benefit the optimization of the warmup regularizers if added, as discussed in  Line 289~304. Also, it is inferior to using our warmup regularizers without BN, because in this case where the variance of each layer is not normalized, we empirically find that the tightness regularizer tends to make the variance of each layer small, to trivially make the value of the tightness regularizer also small. In contrast, when BN is added, the variance of each layer is normalized, and thereby the regularizers can make the bounds tight while the variance remains stable. In other words, our regularizers are mostly effective when BN is used.
>
> | IBP Initialization | Regularizers | BN   | Standard error (%) | Verified error (%) |
> | ------------------ | ------------ | ---- | ------------------ | ------------------ |
> | No                 | No           | No   | 57.08±0.29         | 69.43±0.28         |
> | Yes                | No           | No   | 56.91 ± 0.91       | 69.40 ± 0.44       |
> | Yes                | Yes          | No   | 57.36 ± 0.45       | 69.91 ± 0.31       |
> | Yes                | Yes          | Yes  | 51.72±0.40         | 65.58±0.30         |
>
> ## About related works mentioned by the reviewer
> We thank the reviewers for suggesting these two related works. They are very recent works that also aim to improve the performance of certified robust training. We will cite them and discuss the difference in our revision. They focus on studying why tighter convex/linear relaxation bounds (e.g., Zhang et al., 2020 (CROWN), or Wong et al., 2018) usually cannot outperform IBP using looser interval bounds:
>
> * Lee et al. 2021 identify a factor about the smoothness of loss landscape, and they propose to use tighter bounds via optimization which may lead to more favorable loss landscapes.
> * Jovanovic et al., 2021 identify two properties of convex relaxations, continuity and sensitivity, that may impact training dynamics.
>
> While they focus on improving tighter convex/linear relaxation based training, we focus on improving IBP training which is inherently more efficient. We further mitigate two key issues about bound tightness and ReLU balance, and we reduce the training schedule, for fast IBP training.
> In terms of improving verified errors, Jovanovic et al., 2021 only have preliminary results on a small network for MNIST since their new relaxations require solving convex/linear programs. And we outperform Lee et al., 2021 (15.42% for MNIST eps=0.4; 69.70% for CIFAR-10 eps=8/255) with a notable margin (we get much lower verified errors 10.82% on MNIST and 65.03% on CIFAR-10, while using short training schedules).

---

### Decision · Program_Chairs · 2021-09-28

**Decision:**

Accept (Poster)

**Comment:**

This paper makes useful contributions to robust training. The authors begin by identifying several issues with current certified robustness training and next propose effective approaches to mitigating these issues. The improvement is shown to be substantial in their experiments.

**Consistency Experiment:**

NeurIPS has a long history of experimentation. In 2014, NeurIPS ran an experiment in which 10% of submissions were reviewed by two independent committees to quantify the randomness in the review process. This year, we repeated a variant of this experiment to see how the quality of the review process has changed over time.  This paper was part of the experiment and was therefore assigned to two committees (consisting of reviewers, an Area Chair, and a Senior Area Chair) that reached independent decisions.  If both committees made the same recommendation, this recommendation was followed. If a single committee recommended acceptance, the paper was accepted (with the exception of a few cases in which the other committee identified what we considered a fatal flaw, e.g., an error in a key result).

Both committees reached the same decision: **Accept (Poster)**

The other committee assigned to the paper recommended **Accept (Poster)**.  You can find the other set of reviews, along with any follow up discussion with the authors here:
https://openreview.net/forum?id=AQ9UL-7UvZx